



# Warm conveyor belts in present-day and future climate simulations. Part I: Climatology and impacts

Hanna Joos [1*], Michael Sprenger [1*], Hanin Binder [1], Urs Beyerle [1], and Heini Wernli [1]

[1]Institute for Atmospheric and Climate Science, ETH Zurich, Zurich, Switzerland
[*]These authors contributed equally

**Correspondence:** Hanna Joos (hanna.joos@env.ethz.ch)

**Abstract.** This study investigates how warm conveyor belts (WCB) will change in a future climate. WCBs are strongly ascending airstreams in extratropical cyclones which are responsible for most of their precipitation. In conjunction with the strong cloud formation, latent heat is released which has an impact on the potential vorticity distribution and therefore on the atmospheric circulation in the mid- and upper-troposphere. Because of these and other impacts of WCBs, it is of great importance

to investigate their changes in a warmer climate. To this aim, future climate simulations (RCP8.5 scenario; 2091-2100) are performed with the Community Earth System Model version 1 (CESM1) and compared to a present-day climate (1991-2000). WCB trajectories are calculated based on the six-hourly 3D wind fields. WCBs are represented reasonably well in terms of location and occurrence frequency compared to WCBs in the ERA-Interim data set. In a future climate, WCB inflow regions in the North Pacific are systematically shifted northward in winter, which is in agreement with the northward shift of the storm

track in this region. In the North Atlantic, increased frequencies are discernible in the southwest and a decrease to the south of Iceland. Finally, in the Southern Hemisphere, WCB frequencies increase in the South Atlantic, whereas they decrease near Madagascar. Part of these changes are consistent with corresponding changes in the occurrence frequencies of extratropical cyclones, i.e., the driving weather systems of WCBs. Changes are also found in the WCB characteristics, e.g., in specific humidity of the WCB inflow, the WCB-related precipitation, the cross-isentropic ascent and the isentropic level reached by the

WCB outflow. This has implications for WCB impacts in a future climate. For instance, the strong increase in inflow moisture leads to: (i) a strong increase in WCB-related precipitation, especially in the upper percentiles, thus extreme precipitation related to WCBs might increase; (ii) a strong increase in diabatic heating in the mid-troposphere; and (iii) a higher outflow level which favours WCBs to more strongly interact with the upper-level Rossby waveguide. In summary, by investigating a distinct weather system, the WCB, and how it changes in its occurrence frequency and characteristics in a future climate, this study

provides new insights into the dynamics and impacts of climate change in the mid-latitudes.



## 1 Introduction

The two-way interaction between clouds and the large-scale atmospheric circulation constitutes one of the grand challenges in understanding the Earth's climate system and its accurate representation in numerical models (Bony et al., 2015). The specifics of this challenge differ between climate regions since they are characterized by different weather systems and predominant cloud structures. Whereas deep convective clouds dominate in the tropics associated with the Hadley-Walker circulation, shallow clouds prevail in the quasi-permanent subtropical anticyclones (Bony et al., 2015). The extratropical storm track regions, i.e., the regions with the frequent passage of extratropical cyclones and their attendant fronts (Schemm et al., 2018), typically reveal elongated cloud bands, so-called warm conveyor belts (WCBs) (Browning and Emanuel, 1990). Their formation is mainly related to the process of baroclinic instability, i.e., the interaction of near-tropopause Rossby waves with the low-level baroclinic zone (Hoskins et al., 1985). More specifically, the propagation and evolution of Rossby waves, i.e., of upper-level troughs and ridges, determines the regions of dynamically forced ascent (Davies, 2015) and thereby the formation of clouds and precipitation. However, the formation of clouds can also feed back on the circulation, via both radiative effects of clouds and the release of latent heat during condensation and freezing in clouds. Both radiation and latent heat release lead to diabatic changes of potential temperature and, except for the tropics, of potential vorticity, which in turn affect the atmospheric flow (Hoskins et al., 1985; Stoelinga, 1996). A detailed investigation of these linkages between clouds and the circulation, based on observations, reanalyses and model simulations, is essential for improving the capability of climate models to realistically simulate the observed climate and to provide useful climate projections for a warmer future (Bony et al., 2015; Shaw et al., 2016). This study aims at contributing to this endeavour by analyzing the occurrence of WCBs, i.e., of a particular cloud system in the extratropical storm track regions in present-day and future climate simulations.

In this paragraph, we introduce the concept of WCBs in more detail, and in particular emphasize their key role in the formation of clouds and precipitation in the extratropics, their associated radiative forcing, and their direct impact on the atmospheric circulation. In satellite imagery, WCBs correspond to stratiform cloud bands with a length of up to 3000 km, often intersected by embedded convection (Browning et al., 1973; Bader et al., 1995; Oertel et al., 2019). These cloud bands correspond to coherent airstreams, which move rapidly poleward while ascending from the boundary layer in the warm sector of an extratropical cyclone to the upper troposphere (Wernli and Davies, 1997). During this saturated ascent, temperatures decrease from typically +20°C to below –50°C, and consequently WCB-related clouds evolve from liquid clouds at the beginning of the ascent at lower latitudes to mixed-phase and eventually to ice clouds in the cold upper troposphere at higher latitudes (Joos and Wernli, 2012; Wernli et al., 2016). WCBs occur most frequently in winter (Madonna et al., 2014), when baroclinicity and the dynamical forcing for ascent are strongest. They contribute essentially to the precipitation climatology in the storm track regions and, in particular, to the formation of extreme precipitation events (Browning and Emanuel, 1990; Pfahl et al., 2014). Embedded convection in WCBs can lead to local maxima in precipitation intensity (Oertel et al., 2019). The net cloud radiative forcing of WCBs at the top of the atmosphere varies strongly along the ascent, from negative values in the equatorward part of the associated cloud band to larger, and in winter positive, values further poleward (Joos, 2019). Overall, WCBs contribute up to 10 W m$^{-2}$ to the climatological winter maximum of net cloud radiative forcing in the central North Atlantic (Joos, 2019),





indicating their important role also in the Earth's radiative balance in the extratropics. Last but not least, WCBs can have a direct effect on atmospheric dynamics. The latent heating in WCBs can contribute to the intensification of the associated extratropical cyclone (Binder et al., 2016) and the downstream upper-level ridge (Grams et al., 2011; Madonna et al., 2014). The latter process implies that cloud bands formed by the poleward ascent of WCBs can significantly affect the amplitude and

propagation of Rossby waves, which in turn determine the downstream flow evolution. As a consequence, a realistic representation of WCBs in models is essential for medium-range weather prediction (e.g., Grams et al., 2018; Rodwell et al., 2018) as well as for climate simulations.

This study investigates, for the first time, the representation of WCBs in simulations with one of the state-of-the-art climate models, the Community Earth System Model (CESM, see section 2). More specifically, we address the following research

questions:

- Is CESM, in present-day simulations, able to reasonably capture geographical patterns and seasonal frequencies of WCBs, as compared to the corresponding climatologies in ERA-Interim?

- How do these geographical distributions and frequencies change in a future climate, and how do these changes relate to corresponding signals in the WCB-driving extratropical cyclones?

- How do the characteristics of WCBs, like inflow moisture, associated precipitation, total diabatic heating and the isentropic outflow level change in a warmer climate?

- What are the implications of the geographical changes and the changed characteristics for the atmospheric circulation and precipitation?

A particularly challenging aspect is the identification of WCBs from large climate data sets. The established procedure to

identify WCB airstreams is based on Lagrangian air parcel trajectories (e.g., Wernli and Davies, 1997; Madonna et al., 2014). To compute accurate trajectories, six-hourly three-dimensional wind fields on all model levels are required, which are typically not available from climate model simulations. Therefore, this study is based on re-simulations of the CESM large ensemble (Kay et al., 2015) and the storage of the three-dimensional fields every six hours.

The presented study is organised as follows: The technical details of the WCB/cyclone identification are explained in Section

2, and Section 3 presents WCB frequencies in present-day climate CESM simulations and a comparison with the WCB climatology based on ERA-Interim reanalyses (Madonna et al., 2014). Section 4 then reveals the effects of climate change on the frequency and location of WCBs, according to RCP8.5 scenario simulations with CESM, and Section 5 discusses the potential impact of these WCB changes, e.g., on surface precipitation, mid-tropospheric diabatic heating and upper-level disturbances. Finally, Section 6 summarizes the main results, provides some caveats of the study and presents an outlook.



## 2 Data and Methodology

### 2.1 CESM simulations

The present-day and future climate simulations are performed with the Community Earth System Model version 1 (CESM, Hurrell et al., 2013), based on restart files from the CESM large ensemble project (CESM-LENS, Kay et al., 2015). Atmo-
spheric fields are saved at six-hourly temporal resolution, with a horizontal resolution of approximately 1° and 30 vertical levels. For the period 1990–1999 (present-day climate) and 2091-2100 (end-of-century climate) five ten-year members of the ensemble will be used to identify WCBs, resulting in 50 years of CESM data for present-day climate (1990-1999; HIST) and 50 years for an end-of-century climate (2091-2100; RCP85). In particular, the WCB identification is based on the following fields of the CESM simulation: horizontal wind components (U/V, in m s$^{-1}$), vertical wind speed ($\omega$, in Pa s$^{-1}$) and surface
pressure (in hPa). Additionally, mean-sea level pressure (PSL, in hPa) is used to identify surface cyclones.

### 2.2 WCB Identification

Warm conveyor belts are identified in the CESM simulations in the same way as for ERA-Interim (Dee et al., 2011) in Madonna et al. (2014) and publicly made available (Sprenger et al., 2017). Basically, 48-hour forward trajectories are started every six hours in the 50-year present-day and future-climate period from a global grid with 80 km equidistantly spaced starting points.
Vertically, the air parcels are released from 14 equidistant levels spanning the range from 1050 hPa to 790 hPa. If starting positions on these isobaric surfaces fall below the topography, they are neglected. Typically, around 680'000 trajectories are started every six hours. The kinematic trajectories are calculated with the Lagrangian Analysis Tool Lagranto (Wernli and Davies, 1997; Sprenger and Wernli, 2015), based on the six-hourly CESM wind fields. If trajectories intersect the topography, which can occur because of numerical inaccuracies or unresolved wind components in the CESM simulations, the air parcels
are lifted to 10 hPa above topography and allowed to continue their path.

Based on this global set of 48-hour forward trajectories, potential WCB trajectories are identified by application of a simple ascent criterion: the trajectories must ascend at least 600 hPa from their release in the lower troposphere until their arrival in the upper troposphere, 48 h later. All trajectories fulfilling this criterion are accepted as potential WCB trajectories, all the others – a huge majority – are neglected (see Figure 1). A more technical problem refers to the end-of-year transition of the trajectories.
In fact, the CESM simulations come in blocks of 10-year continuous periods, but we treat all the years in the 50-year periods separately. Therefore, the trajectories are not allowed to cross the year's end, with the implication that four days around the end of the year are not covered in this WCB climatology.

Two further selection criteria are subsequently applied to the potential WCB trajectories. First, we make use of the conceptual view of the WCB as a coherent airstream attributed to an extratropical cyclone. To this aim, surface cyclones are identified
in the sea-level pressure fields for the CESM simulation with the method by Wernli and Schwierz (2006) and further refined in Sprenger et al. (2017). The resulting cyclone masks, defined as the outermost closed isobar surrounding a local pressure minimum, are then required to be intersected by any (real) WCB trajectory. To exclude trajectories that might be ascending in a tropical cyclone, we artificially set the cyclone masks to zero in a tropical band between 25°S and 25°N. The intersection



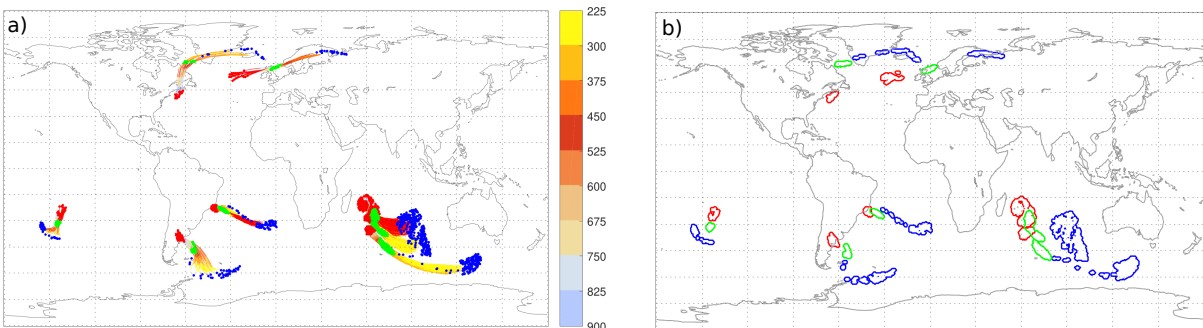

**Figure 1.** Example of identified WCB trajectories started on 00 UTC 15 December 1990 in an ensemble member of HIST. (a) 48-hour forward trajectories, colored according to pressure (in hPa). The time instances 0 h (start of ascent), 24 h (during ascent) and 48 h (end of ascent) are marked with red, green and blue circles, respectively. (b) WCB masks, with a 100 km blow-up radius, at the same time instances. The three WCB masks at times 00 h, 24 h and 48 h, averaged over the whole ERA-Interim and CESM periods and ensemble members, build the basis for the climatologies in Section 3 and 4 (see text for details).

of a WCB trajectory with a cyclone mask might occur at any time instance during the 48-hour ascent period. In addition to only be used as a WCB-selecting feature, the cyclone frequencies in CESM are – of course – by itself of great interest, and will be discussed also as a distinct meteorological feature in conjunction with the WCBs. Note that the attribution of potential WCB trajectories to cyclones relies on all cyclone masks, irrespective of the lifetime of the cyclone. In addition, we apply a

cyclone tracking algorithm to follow the cyclones from genesis, to maturity and finally lysis. Based on this cyclone tracking, we also apply a lifetime filter of 24 hours to the cyclone tracks, and in the discussion of cyclone climatologies, we restrict the analysis to the set of these more relevant systems. For the WCB/cyclone attribution, however, we keep the full cyclone masks in order not to lose WCB trajectories by a too restrictive cyclone criterion. The resulting trajectories all fulfill the required ascent criterion of a WCB and are tied to a (extratropical) cyclone. A second selection criterion guarantees that the trajectories started

at consecutive (every six hours) time instances are not counted multiple times. A detailed description of this double-count (or multiple-count) filter, illustrated by specific examples, can be found in Madonna et al. (2014). An example of the selected WCB trajectories is shown in Figure 1a, including the 48 h trajectories and the position of the air parcels along the ascending airstreams at the times t=0 h (red dots), t=24 h (green dots) and t=48 h (blue dots).

     Finally, the Lagrangian WCB trajectories are gridded to a regular latitude/longitude grid with 0.5°x 0.5° horizontal resolution

(see Figure 1b). These gridded 2D fields build the basis for the climatological analysis of WCBs in CESM. More specifically, at every six-hour timestep it is determined if a grid point is nearby (< 100 km) a WCB trajectory. We do that separately for all times of the WCB ascent, yielding maps for the location of WCB air parcels at the times t=0 h, 6 h, ... 48 h.

## 2.3   Test for statistical significance

Geographical maps of yearly and seasonal WCB/cyclone frequencies will be presented in this study, for ERA-Interim, HIST

and RCP85. If we assume that HIST realistically captures atmospheric dynamics and thus provides a set of 50 WCB/cyclone



maps that represent present climate, two questions arise. First, how do the 37 yearly and seasonal WCB/cyclone maps from ERA-Interim (1979-2016) compare to the 50 years of HIST simulations and corresponding yearly and seasonal WCB/cyclone maps? Second, how do the 50-year HIST and RCP85 simulations, and the corresponding WCB/cyclone maps compare to each other? In summary, these two questions would ask for a careful statistical analysis to see if the differences between

ERA-Interim/HIST and HIST/RCP85 are statistically significant, and thus point to systematic differences or reflect only the natural variability. However, given that we only have $2 \times 50$ years of climate simulations and 37 years of ERA-Interim data at hand, and we therefore can only have, e.g., the same number of seasonal WCB/cyclone maps for comparison, a robust statistical inference remains elusive. A robust statistical analysis would ask for about ten times more maps, which however is not feasible due to the computational constraints (in particular, the large computational cost of trajectory calculations). The

ERA-Interim/HIST and HIST/RCP85 comparisons will therefore remain at a rather qualitative level, still relying however on physical plausibility.

In contrast to the seasonal and yearly WCB/cyclone maps, the 50 and 37 years for CESM and ERA-Interim, respectively, contain a large number (many thousands) of single WCB events. This allows for a statistically robust comparison of key characteristics (e.g., diabatic heating rates, precipitation) of these events. The adopted statistical approach to analyse potential

shifts in the characteristics will be introduced and discussed in Section 4.

## 3 Representation of WCBs in CESM present-day climate

In this chapter we discuss the representation of WCBs in the HIST simulation and compare the results to the WCB climatology based on the ERA-Interim data set (Madonna et al., 2014).

In Figure 2 the frequency of occurrence of WCBs in boreal winter (DJF) is shown for ERA-Interim (left column) and HIST

(right column). WCBs are most frequent in the extratropical storm tracks in the winter hemisphere, as they are per definition linked to extratropical cyclones which in turn constitute the storm tracks and are most prevalent during winter. Two maxima in the WCB starting (or inflow) regions, i.e., the locations where the WCBs start their 48 h ascent, are discernible in the North Atlantic and North Pacific (see black lines) as well as in a band encircling the Southern Ocean between 25°S and 40°S. During their main mid-tropospheric ascent, WCB air parcels travel east- and poleward leading to frequency maxima of $\sim 7\%$ (5%) in

the Northern (Southern) Hemisphere in the storm track regions (see colours). The WCB outflow, i.e., the regions covered by WCB air parcels at the end of their 48-hour ascent, is spread out over large parts of the extratropical hemispheres (red lines). During boreal summer (JJA; Figure 2c,d), two maxima in WCB starting and ascent regions are discernible over the North American continent and its east coast as well as over the Northeast Pacific and the Himalayan region. Whereas the latter is connected to the Asian monsoon, the maximum over the Northeast Pacific is linked to the Mei-yu-Baiu front. A more detailed

description of the WCB climatology, i.e., its regional hotspots and seasonal cycles, can be found in Madonna et al. (2014).
Comparing the WCB climatology in ERA-Interim (Figure 2, left columns) to the one calculated based on HIST (right column), it is striking to see the similarity between the two, which thus points to CESM's capability to realistically simulate WCBs. In fact, both the frequency amplitude and the geographical location of their starting, ascent and outflow regions agree very

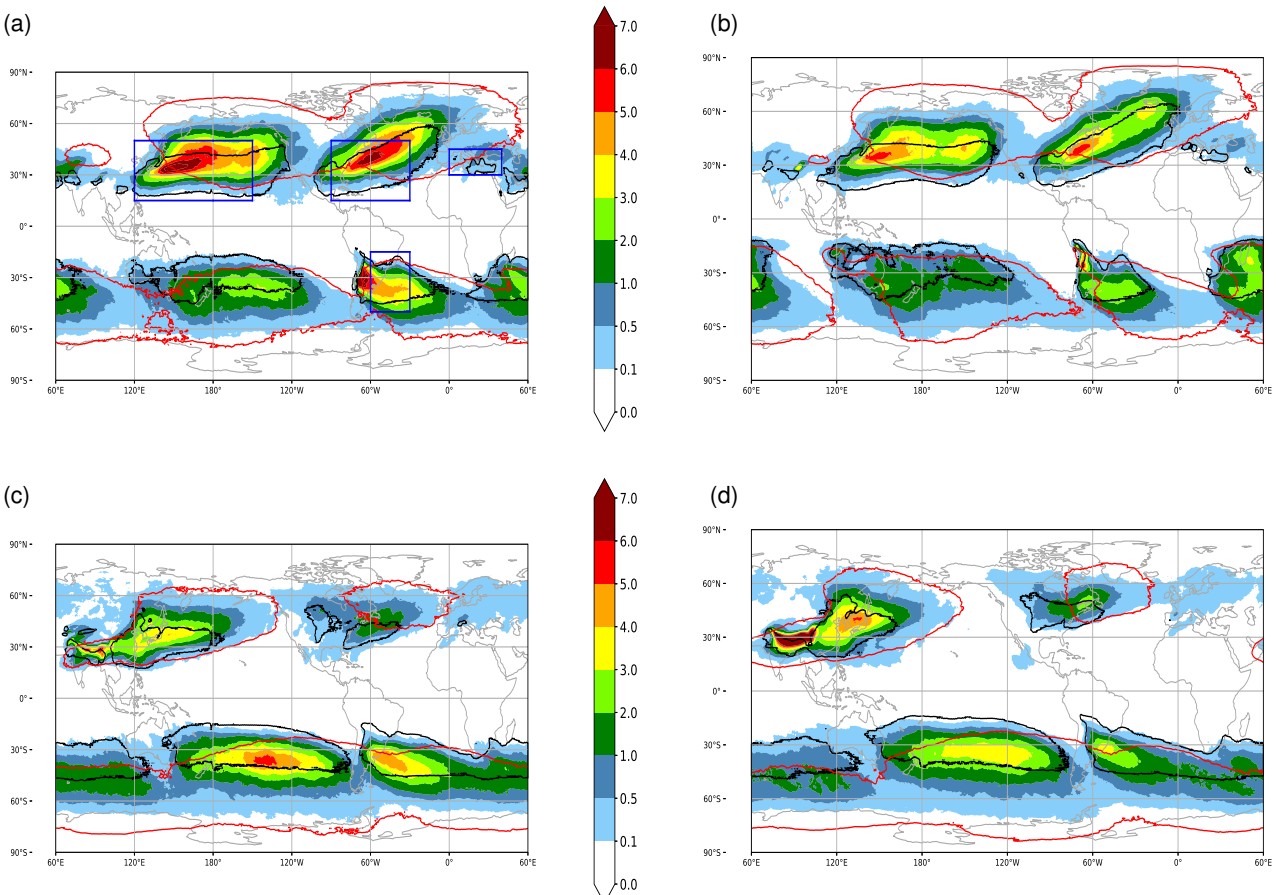

**Figure 2.** Climatological frequency of WCBs for DJF (a, b) and JJA (c, d) for ERA-Interim (a, c) and HIST (b, d). Colours denote the percentage of six-hourly timesteps during which at least one WCB trajectory is located in a circle of 100 km around the considered grid point, 24 h after the start of the ascent. The black line shows a frequency of 1% for WCBs at their starting time t=0 h and the red line shows a frequency of 1% for WCBs at the end of their ascent at t=48 h. The blue boxes in panel (a) show the regions which are used for the regional analysis of WCBs in Figures 8 and 9.

well between both data sets. Of course, regional differences can still be identified, for instance, in the poleward and eastward movement of WCBs (e.g., in the North Atlantic and North Pacific) and also in the frequency amplitudes (e.g., in the lee of the Andes and the Himalayas). Whereas the maps in Figure 2 show that HIST agrees well with ERA-Interim in capturing the frequencies with which a region is affected by a WCB, they do however not quantify the 'intensity' of the ascending airstreams, which most easily is defined as the number of ascending WCB trajectories per six-hour time interval. This information is listed in Table 1 for ERA-Interim and HIST, and for later reference also for RCP85. The numbers for ERA-Interim and HIST agree very well, in particular during winter in the Northern and Southern Hemisphere. During summer, HIST seems to overestimate the number of WCB trajectories, in particular in the Northern Hemisphere. Some additional information on the WCB intensity




**Table 1.** Number of WCB trajectories per six-hour interval for ERA-Interim, HIST and RCP85 determined as an average for the 37 ERA-Interim and 50 CESM years. The different columns give: the global and yearly average (column 2), the winter averages for the Northern and Southern Hemisphere (columns 3,4), and correspondingly the summer averages (column 5,6).

| | all seasons | NH winter (DJF) | SH winter (JJA) | NH summer (JJA) | SH summer (DJF) |
|---|---|---|---|---|---|
| ERA-Interim | 447 | 298 | 305 | 108 | 189 |
| HIST | 476 | 271 | 302 | 174 | 219 |
| RCP85 | 601 | 325 | 372 | 263 | 269 |

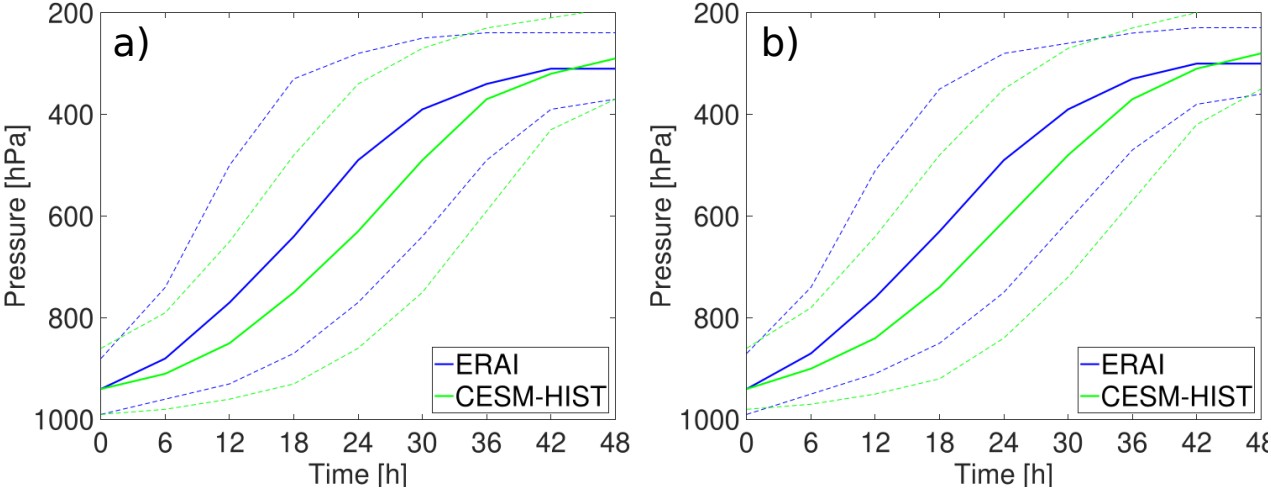

**Figure 3.** Mean time evolution of pressure (in hPa) along all WCB trajectories during (a) winter and (b) summer. Solid lines denote the mean over all trajectories and dashed lines the 10th and 90th percentiles for ERA-Interim (blue) and HIST (green).

can be gained from intensity maps, which complement the frequency maps in Figure 2 and are available in the Supplementary Material (Figure S1 and S2).

In addition to the geographical pattern of WCB occurrence, we also analysed the ascent behaviour in both data sets. In Figure 3, the time evolution of pressure along the 48-hour ascent for all WCB trajectories is shown. During both seasons, WCBs

5   start from the lower troposphere at ∼ 950 hPa and ascend to a height of ∼ 300 hPa, in the mean. The average time WCB air parcels in HIST need to fulfill the 600-hPa ascent criterion equals to ∼ 42 h. Thereby, the ERA-Interim trajectories reach this criterion slightly earlier than the HIST trajectories, which is also reflected in the faster and earlier ascent in ERA-Interim. However, given the differences in (vertical) resolution between ERA-Interim and CESM (see section 2.1) the ascent behaviour is remarkably similar between the two models.

10   In summary, the agreement between ERA-Interim and HIST suggests that the latter is able to simulate, for a present-day



climate, the WCBs' spatial occurrence, their frequency and ascent behaviour in a physically reasonable way (see comments in section 2.3 about statistically robust inferences). In fact, an exact agreement cannot be expected because two models are compared that substantially differ in their parameterizations of sub-grid scale processes and their (vertical) resolutions. Furthermore, two different time periods are included in both data WCB sets. Whereas for calculating the ERA-Interim trajectories,

the time period from 1979-2018 is used, for HIST we use five times ten years representative for 1990-2000 to get a 50-year climatology. It is not a priori clear how ERA-Interim's single realisation of the present-day climate 'fits' into the 1990-2000 spectrum of CESM's present-day climate (see also section 2.3). The results of this section, however, point to a physically reasonable representation of WCBs in HIST, and therefore we will use HIST in combination with RCP85 to assess the potential change of WCBs in a future climate in Section 4.

## 10  4  WCBs in a Future RCP8.5 Climate

In this section, WCBs in HIST will be compared to the ones in RCP85. To this aim, we calculated WCB trajectories in five ensemble members of the RCP85 simulation that are representative for the years 2090-2099 under the RCP8.5 emission scenario (see Section 2.1 and 2.2). In a first part (section 4.1) the focus will be on differences between the geographical patterns and occurrence frequencies. Then, in section 4.2, we will characterize the WCBs in several hotspot regions by means of key

parameters (e.g., their associated precipitation), followed by a more refined analysis of a potential shift to more extreme values.

### 4.1  Geographical changes in WCB occurrence frequencies

The absolute frequencies of WCBs in HIST as well as the difference in the WCB frequencies between HIST and RCP85 are displayed in Figure 4 for DJF and for the times t=0 h (starting locations), t=24 h (ascent) and t=48 h (outflow) separately. In a future climate, more WCBs start near the U.S. East Coast and over the U.S. Southern States (Figure 4b). On the other hand,

the starting frequencies to the south of Iceland are decreased. In the North Pacific, the opposite signal can be observed, with a decreased frequency in a future climate around 30°N and an increased frequency further to the north. Climate-related changes are also discernible in the Southern Hemisphere. For instance, over South America, the WCB frequency at time t=0 h is slightly increased, whereas it is somewhat decreased near Madagascar.

In the following 24 h, i.e. during the WCBs ascent (Figure 4c,d), the differences observed in the starting frequencies move

downstream (to the east) leading to decreased frequencies over the North Atlantic around Iceland, around 30°N in the Pacific and to the east of Madagascar. An increased WCB frequency in RCP85 is also discernible near the U.S. East Coast, to the north of 30°N in the North Pacific and to the east of South America.

After 48 h and thus at the time when the trajectories reach upper-tropospheric levels (Figure 4e,f), the WCB trajectories spread out substantially and, consequently, the differences between HIST and RCP85 cover considerably larger areas than at time

t=0 h or t=24 h. In the North Atlantic southward of 60°N, higher frequencies are found in RCP85, whereas lower frequencies prevail northward of 60°N. In contrast, over the North Pacific an opposing signal is found: the WCB outflow is less often located around 30°N but, in accordance with the northward shift of the WCB inflow (t=0 h), while an increased frequency can



be seen between 45°N and 60°N. Finally, in the Southern Hemisphere, the difference signal at time 24 h is also propagating further downstream and at time t=48 h covers considerably larger areas due to the spreading out of the WCB airmasses in the upper-tropospheric outflow.

The differences between HIST and RCP85 for the Northern Hemispheric summer (JJA) are displayed in Figure 5. Over the
Central U.S. and East Canada, a dipole is discernible: fewer WCBs start over the Central U.S. in RCP85, but more WCBs start over East Canada. In the Himalayas and the adjacent Pacific Ocean, the most pronounced increase in WCB frequency is observed in RCP85. Also the Southern Hemisphere exhibits higher frequencies over the Atlantic and Indian Ocean. On the other hand, a decrease over the West Pacific is observed. These differences in the WCB starting regions (t=0 h) again propagate downstream with time (see Figure 5d). At time t=48 h (see Figure 5f) a considerable increase in RCP85 is discernible in large
areas over South East Asia and the West Pacific as well as the Southern Ocean.

The maps in Figures 4 and 5 compare the *frequencies* with which a specific region is affected by a WCB. However, they neglect – as in the comparison between ERA-Interim and HIST in Section 3 – a potential change in WCB *intensities*. Indeed, a substantial increase in WCB intensity is discernible in a warmer climate, by again comparing the number of WCB trajectories per six-hour interval (see Table 1). Consistently, the numbers are higher (by about 26%) in RCP85 than in HIST for all seasons
and separately in the Northern and Southern Hemisphere. This increase is also seen in the WCB-intensity maps provided in the Supplementary Material (Figure S1 and S2).

In section 3 we compared the ascent pressure evolution of WCB trajectories in ERA-Interim and HIST, finding that the overall ascent is well captured in CESM in particular with respect to the maximum altitude reached by the WCB air parcels. Here, we now compare the ascent behaviour of WCB trajectories in HIST and RCP85 (Figure 6), expecting that the overall warmer
atmosphere and increased moisture content potentially influence the maximum altitude reached but also the ascent rate of the WCBs. However, the global mean pressure decrease during the 48-hour ascent is very similar in both simulations. In fact, during DJF, trajectories start from the same pressure levels in HIST and RCP85, and they ascend to only slightly higher levels in RCP85. During JJA, the starting and ending pressures are slightly higher in RCP85, but this vertical shift is not accompanied by a pronounced change in the WCB ascent rate, at least in the global and seasonal mean shown in Figure 6. This result
indicates that the WCBs in a future climate keep their essential ascent characteristics (but see also Section 4.3).

Finally, we ask the question how the climate-related shifts in WCB frequencies can be explained. To this aim, it is worthwhile to keep in mind that the WCB is one of the characteristic airstreams in extratropical cyclones, which is also reflected in the fact that our WCB definition (in section 2.2) explicitly links any WCB to a cyclone. This, in turn, means that any shift in WCB frequencies (spatially and in amplitude) leads to a corresponding question concerning cyclones as the WCB-driving systems. In
fact, several of the signals discussed before can be associated with corresponding signals in cyclone frequencies. For instance, during DJF the WCB frequency in the North Pacific decreases in RCP85 around 30°N and increases further to the north. This signal is consistent with the corresponding change in cyclone frequency in the North Pacific, where fewer cyclones are found around 30°N and more around 45°N (see Figure 7b). Note that a northward shift of the Pacific storm track in a warmer climate is also found in other studies (e.g. Tamarin-Brodsky and Kaspi, 2017; Priestley and Catto, 2022). Consistent signals are also
found over South America, where both the WCB frequency at time t=0 h and the cyclone frequency are slightly increased. A



**Figure 4.** Absolute values of WCB frequency (in %) of HIST (a, c, e) and difference between RCP85 and HIST during boreal winter (DJF) for WCB ascent times t=0 h (b), t=24 h (d), t=48 h (f).

(partly) consistent decrease in WCB and cyclone frequencies is found over Madagascar, where fewer cyclones occur in an area



**Figure 5.** Same as Figure 4 but for JJA.

reaching from Madagascar to the northwest coast of Australia.

There are, however, also regions where cyclone and WCB frequencies do not exhibit consistent signals. For instance, this is



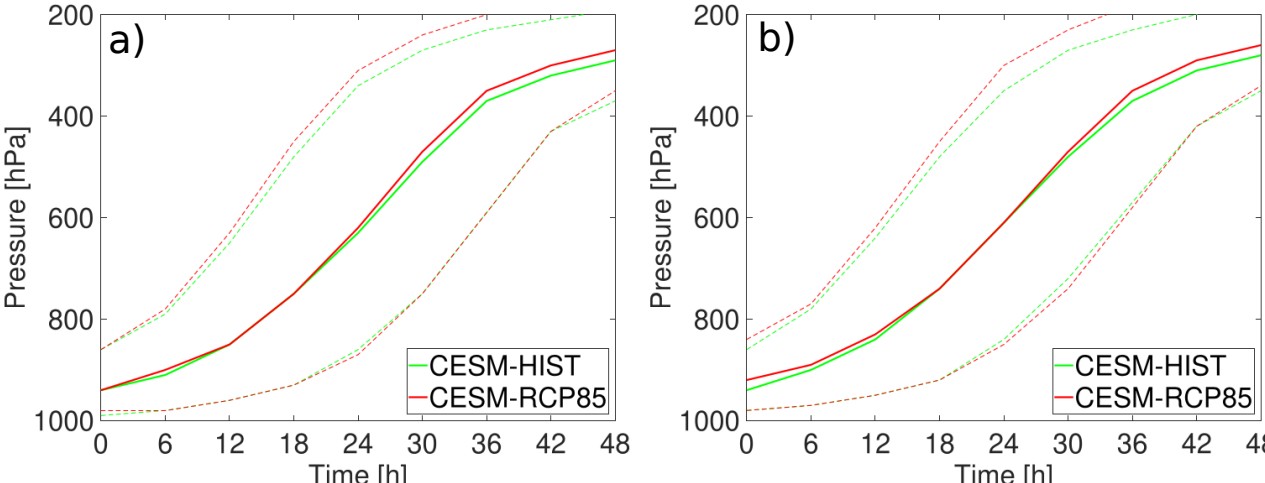

**Figure 6.** Time evolution of pressure (in hPa) along all WCB trajectories in (a) winter and (b) summer. Solid lines denote the mean over all trajectories and dashed lines the 10th and 90th percentiles for HIST (green) and RCP85 (red).

the case in the North Atlantic and during DJF. There, WCBs increase over the western North Atlantic and decrease in the East Atlantic, south of Iceland. The cyclone frequencies, in contrast, completely lack such a clear dipole structure. This discrepancy points to an important aspect: in addition to frequency changes of cyclones in a specific region, it might also be the cyclone structure itself that changes and thus allows for more or less WCBs to occur. Stated otherwise, in a warming climate the WCB-

efficiency of cyclones due to changes in inflow moisture and/or baroclinicity could (regionally) change and thus explain any climate-related WCB signals. The decomposition (or attribution) of the WCB changes into a cyclone frequency and efficiency part is, however, not trivial and beyond the scope of this study.

### 4.2 Regional changes of WCB characteristics

Despite the very similar ascent behaviour in both considered climates (Figure 6), we have shown that differences in the WCB

frequency occur in the main WCB ascent regions (Figures 4,5). Therefore, we now analyse characteristics of WCBs and associated WCB impacts like precipitation separately for the world's main WCB ascent regions, namely the North Atlantic (NATL) and North Pacific (NPAC) storm tracks, the Mediterranean (MED) and the South Atlantic (SATL). The selected regions are shown in Figure 2a. We characterize the WCBs by calculating different measures: the specific humidity at time t=0 h, i.e., at the beginning of the WCB ascent; the accumulated precipitation during the ascent (t=0...48 h); the difference in potential

temperature ($\Delta\theta$) between the start and the end of the ascent[1]; and the potential temperature at the end of the ascent ($\theta_{end}$). The results are shown in Figure 8 for winter only, i.e., DJF for the North Atlantic, North Pacific and Mediterranean and JJA for

---

[1]The end of the WCB ascent is defined as the time when the WCB air parcels no longer increase in altitude, i.e., the pressure increases again. However, to allow for transient phases of descent at lower- to mid-tropospheric levels, the criterion is only applied if the air parcels are at least 400 hPa above their starting altitude. If air parcels continue their ascent during 48 h, the end of the ascent is set to 48 h.



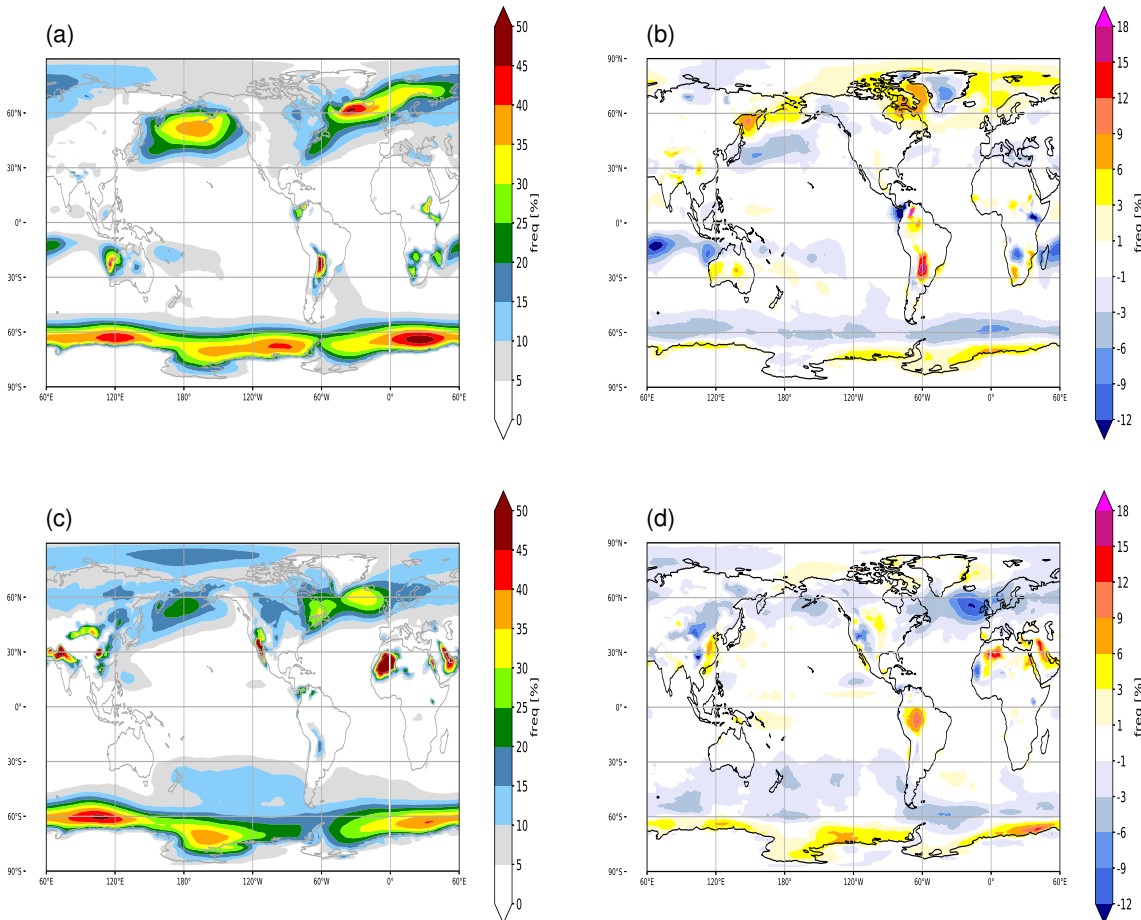

**Figure 7.** Absolute values of cyclone frequency for HIST for DJF (a) and JJA (c) and difference in cyclone frequency between RCP85 and HIST for DJF (b) and JJA (d).

the South Atlantic.

The distribution of specific humidity in the WCB inflow increases substantially from HIST to RCP85 in all considered regions (Figure 8a). The 5th and 95th percentiles as well as the mean and median of the distributions is shifted to significantly higher values, whereas also the width of the distribution is slightly increased in RCP85. The difference is most pronounced in the Mediterranean region where the inner-quartile ranges are almost completely separated. The increased moisture values in the WCB inflow lead in all regions, consequently, to an increase in total precipitation that falls along the ascending airstream (Figure 8c). This increase arises from changes in the large-scale precipitation, whereas convective precipitation remains rather unchanged in RCP85 compared to HIST (not shown). The increase of inflow moisture in a future climate also has a strong impact on the change of potential temperature ($\Delta\theta$), i.e., the cross-isentropic flow along the WCB ascent (Figure 8b). More diabatic heating due to cloud formation leads to the enhanced $\Delta\theta$. This is true for all regions, and it is particularly pronounced



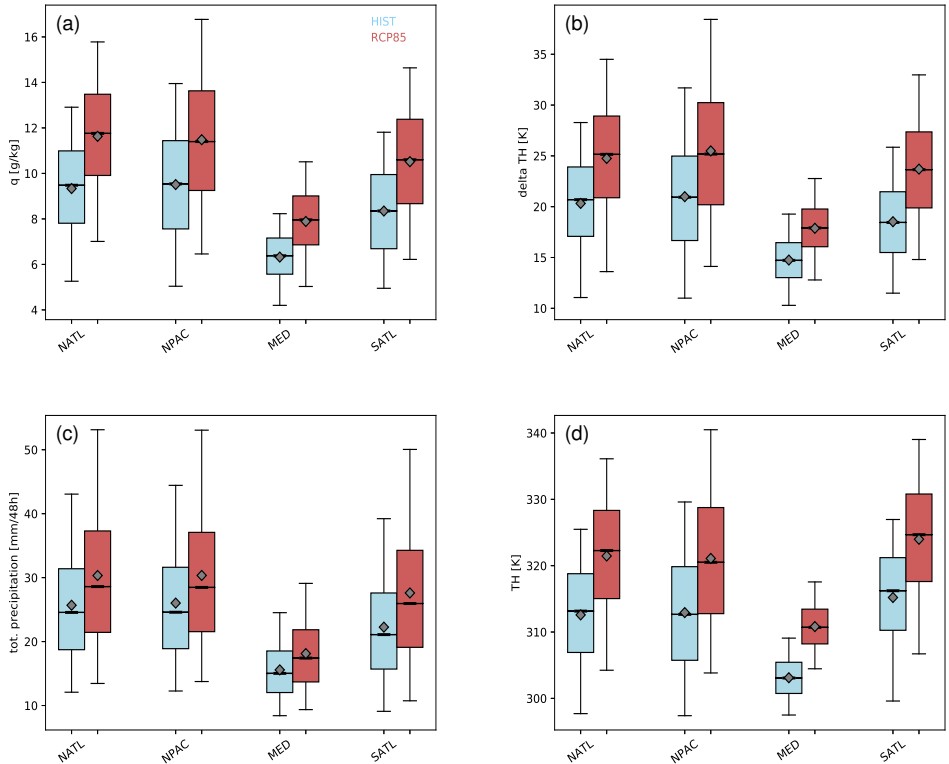

**Figure 8.** Box-and-whisker (BW) plot for different WCB characteristics for the North Atlantic (NATL), North Pacific (NPAC), Mediterranean (MED) and South America (SATL). Only winter is considered, i.e., DJF for NATL, NPAC and MED, and JJA for SATL. The results for HIST are shown in blue bars, for RCP85 in red bars. The BW bars show: lower-to-upper quartile range (colored bar), median (bold black line), mean (diamond), 5th-to-95th percentile range (thin lines).

in the upper percentiles. Connected to the more diabatic nature of WCB ascents and the increase in $\Delta\theta$, WCBs reach higher isentropes at the end of their ascent (Figure 8d). In all regions except for the Mediterranean, WCBs in a warmer climate reach higher than the 320 K isentrope. The difference is most pronounced in the Mediterranean, where the two distributions are almost completely separated for HIST and RCP85. Hence, in a warmer climate WCBs in the Mediterranean reach the 310 K isentrope in the mean, whereas it is only reached by the most extreme WCBs in HIST.

In summer, the properties of WCBs also change in a warmer climate (see Figure 9). Specific humidity in the WCB inflow is increasing in all regions whereby, in contrast to winter, the change is most pronounced in the North Pacific and South Atlantic oceans (see Figure 9a). In the Mediterranean, the smallest change is observed. Due to the increase in moisture, also the precipitation which falls along the ascending WCB airstreams is increased (Figure 9c). The mean/median precipitation only slightly increases in the North and South Atlantic, and almost no increase is discernible in the Mediterranean. The most pronounced increase occurs in the North Pacific in the higher percentiles, where it amounts to an almost 20% increase. Thus extreme precipitation events connected to WCBs potentially will be increased in a future climate in the North Pacific. Interestingly, the

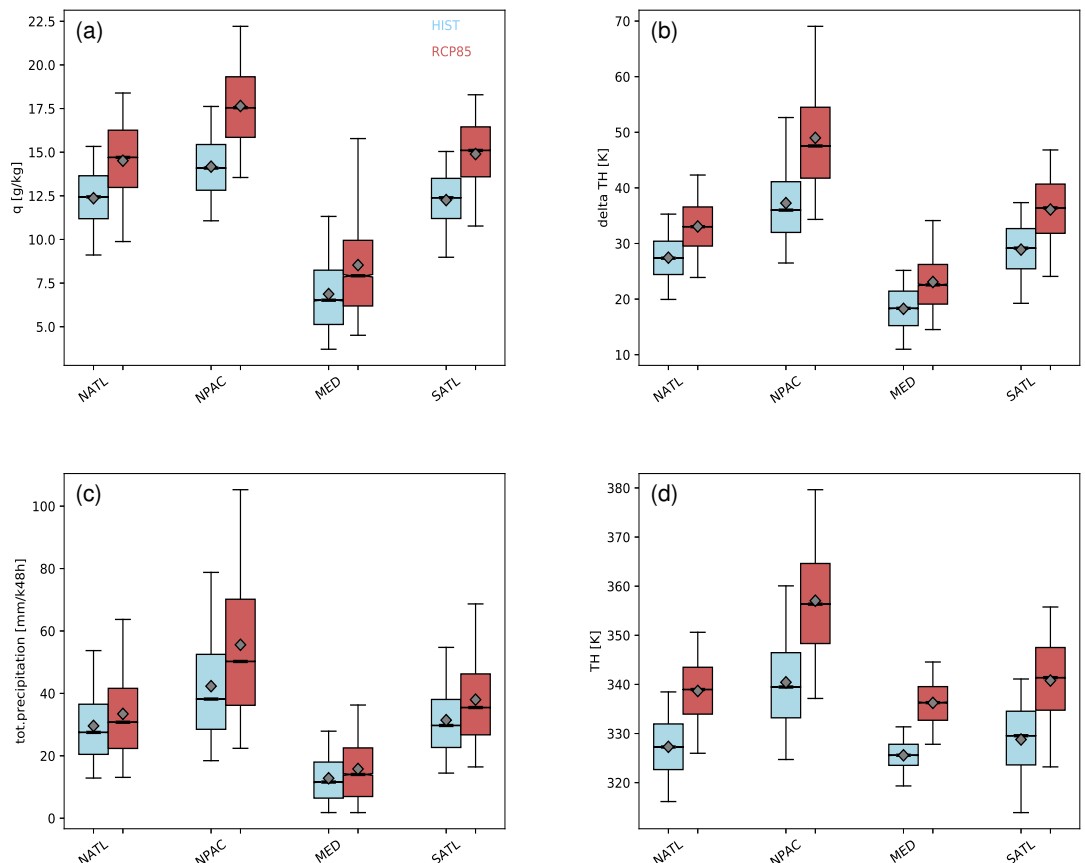

**Figure 9.** Same as Figure 8 but for summer, i.e., JJA for NATL, NPAC and MED, and DJF for SATL.

shift in the upper percentiles is much less pronounced in the North Atlantic. The change of potential temperature is shown in
Figure 9b. Again, the cross-isentropic flow and thus the diabatic character of WCBs is strongly enhanced in a warmer climate.
In all regions $\Delta\theta$ strongly increases whereas the changes are smallest in the Mediterranean region. Linked to the enhanced
diabatic activity, the ascending WCB airstreams reach considerably higher isentropes. Indeed, in all regions the inner-quartile

5 ranges of the distributions are completely separated (see Figure 9d). The increase in the Mediterranean is mainly caused by an
increase in the $\theta$ values at the WCB inflow, which reflects the strong temperature increase in the Mediterranean at the surface
in a warmer climate (not shown).

As mentioned before, the box-and-whisker plots of several characteristics particularly differ in their climate-change-related
shift of the 5th and 95th percentiles. For instance, for precipitation, the former remains rather unchanged, the latter increases

10 substantially in winter from HIST ($\sim 40$ mm/48 h in NATL, NPAC and MED) to RCP85 ($\sim 50$ mm/48 h). Further, in summer,
the upper percentiles of precipitation in the North Pacific exhibit a much more pronounced shift than the corresponding shifts
in the North Atlantic. This indicates that there is actually a shift to more extreme precipitation events in the future climate,
but that it depends on the considered region. This is, however, difficult to quantitatively determine from Figures 8 and 9, and





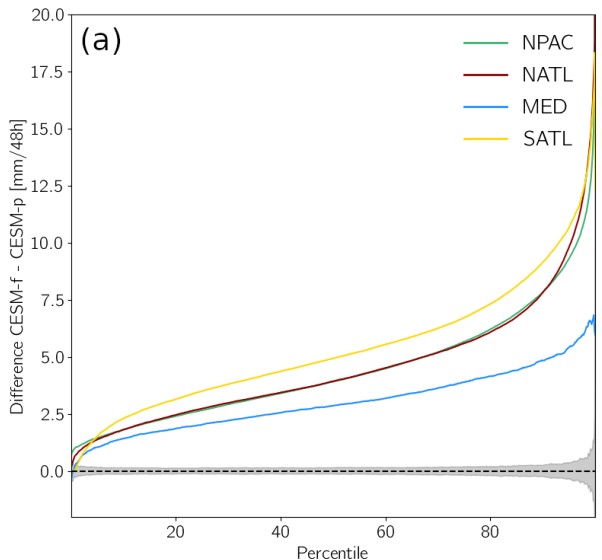 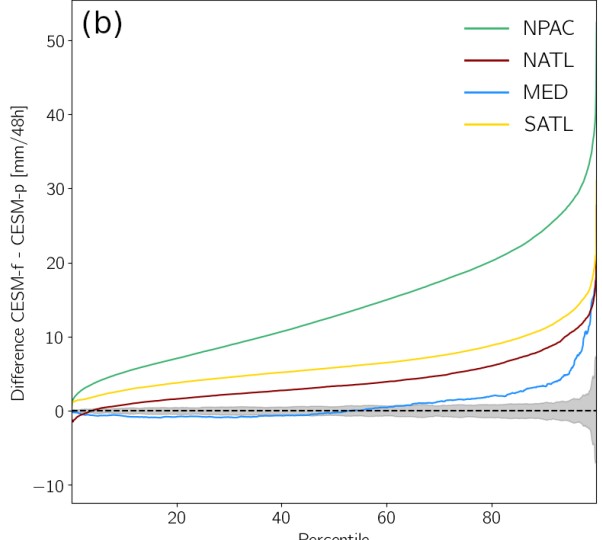

**Figure 10.** Difference of precipitation percentiles for the four target regions, for winter (a) and for summer (b). The grey-shaded area corresponds to a 95% confidence interval under the null hypothesis that the two precipitation distributions for HIST and RCP85 do not differ. Exemplarily, the confidence interval is shown for the region MED; the confidence intervals for the other regions fall within the one for MED. The confidence interval is calculated based on a resampling approach (see text for details). Note the different vertical axis scales in the two panels.

thus we proceed with a more refined statistical analysis that allows for a detailed quantification of the shift in the precipitation percentiles.

In essence, the percentiles of the HIST and RCP85 precipitation are first calculated, and then their difference is determined. It is shown for the four target regions and for winter and summer in Figure 10, i.e., the horizontal axis gives the percentile and the vertical axis the difference in the precipitation at the corresponding percentiles. For instance, the median precipitation (50% percentile) increases between 2.5 mm/48 h (for MED) and 50 mm/48 h (for SATL) in RCP85 compared to HIST. The most striking signal in the percentile differences is, however, the substantial increase towards the more extreme percentiles. In winter the range shifts to 5.0 mm/48 h for MED and around 15.0 mm/48 h for the other regions, which is a very substantial shift to more extreme precipitation. In fact, these shifts can also be compared to the typical precipitation values in Figure 8 and 9. The effect becomes considerably strengthened in summer, particularly in the region NPAC.

To assess the statistical significance of the RCP85/HIST percentile differences in Figure 10, we applied a resampling approach to determine for each percentile difference the 95% confidence interval, i.e., that it significantly deviates from 0. More specifically, we assume that the WCB-related precipitation of HIST and RCP85 does not differ, i.e., belong to a common distribution. Under this null hypothesis, we draw two equal-sized samples and determine the percentiles difference for these two samples. This step is repeated 10'000 times and for each percentile the 2.5%-to-97.5% range of the resampling differences is shaded in





grey. This is exemplarily done for the region MED in Figure 10, and for the other regions the corresponding confidence intervals fall within the MED confidence interval. Hence, if the actual RCP85/HIST percentile for the MED difference falls outside this grey range, it indicates that the null hypothesis is not valid and the difference, at the corresponding percentile, is statistically significant. The fact that this essentially is the case for all percentiles strongly underlines the climate-warming-related shifts in precipitation.

## 5  Implications of changes in WCB characteristics in a future climate

In the previous sections we have shown that WCB frequencies, but also their characteristics considerably change in a warming climate. More specifically, the WCB-related precipitation strongly increases in the upper percentiles, but it remained open where on the globe these changes will be most pronounced. Furthermore, the more diabatic nature of the WCB ascent linked to a higher outflow isentrope has been discussed, whereas the implications of these changes remained unclear. In this section we will therefore discuss in more detail three different WCB impacts that are associated to these changes, namely (i) the interaction of WCB outflows with the upper troposphere / lower stratosphere (UTLS), (ii) mid-tropospheric diabatic heating, and (iii) the geographical distribution of WCB-related surface precipitation.

### 5.1  WCB impact on the upper-tropospheric flow

When WCBs reach the UTLS region at the end of their ascent, they have the potential to modify the upper-level flow and to eventually amplify upper-level ridges, or initiate Rossby waves where the WCB outflow interacts with the PV waveguide (or jet stream) (e.g., Wernli (1997); Pomroy and Thorpe (2000); Grams et al. (2011); Pfahl et al. (2015b)). In consequence, these local modifications of the upper-level flow can propagate downstream and thus influence the evolution and the predictability of the weather downstream (Grams et al., 2011, 2018). For a WCB to exert a substantial impact on the upper-level flow, it must reach levels close to the tropopause. Hence, it is physically most elucidating to show the WCB outflow region together with the position of the (dynamical) tropopause, which we here define as the 2 pvu isoline. This is done in Figure 11, which includes the zonal mean tropopause position, isentropes (potential temperature isolines) as well as the distribution of WCB outflow positions for the four different regions (NATL, NPAC, MED and SATL) and for HIST and RCP85.

The isentropes ascend towards the poles whereas the dynamical tropopause descends from its levels around 100 hPa in the tropics to about 300 hPa in the polar regions, with steep slopes where jet streams are located. Due to the warmer temperatures in RCP85, the isentropes are consistently shifted to lower altitudes in the troposphere (dashed gray lines). The tropopause, however, remains rather unchanged in the warmer climate (solid/dashed black lines). In HIST, air parcels in the WCB outflows, i.e., at the end of their ascent, reach the UTLS in the mean at a latitude of $\sim 45°$N/S (Figure 11) for the oceanic basins (NATL, NPAC and SATL) and of $\sim 50°$N for the Mediterranean (MED) (see green lines). Interesting is the difference in outflow altitude relative to the dynamical tropopause. Whereas the outflow peak is located below the mean tropopause in NATL and MED, this is not the case for NPAC and SATL. There, the maximum of the outflow is either located very near to the tropopause altitude, or even within the lower stratosphere. In the RCP85 climate, all regions experience a poleward and upward



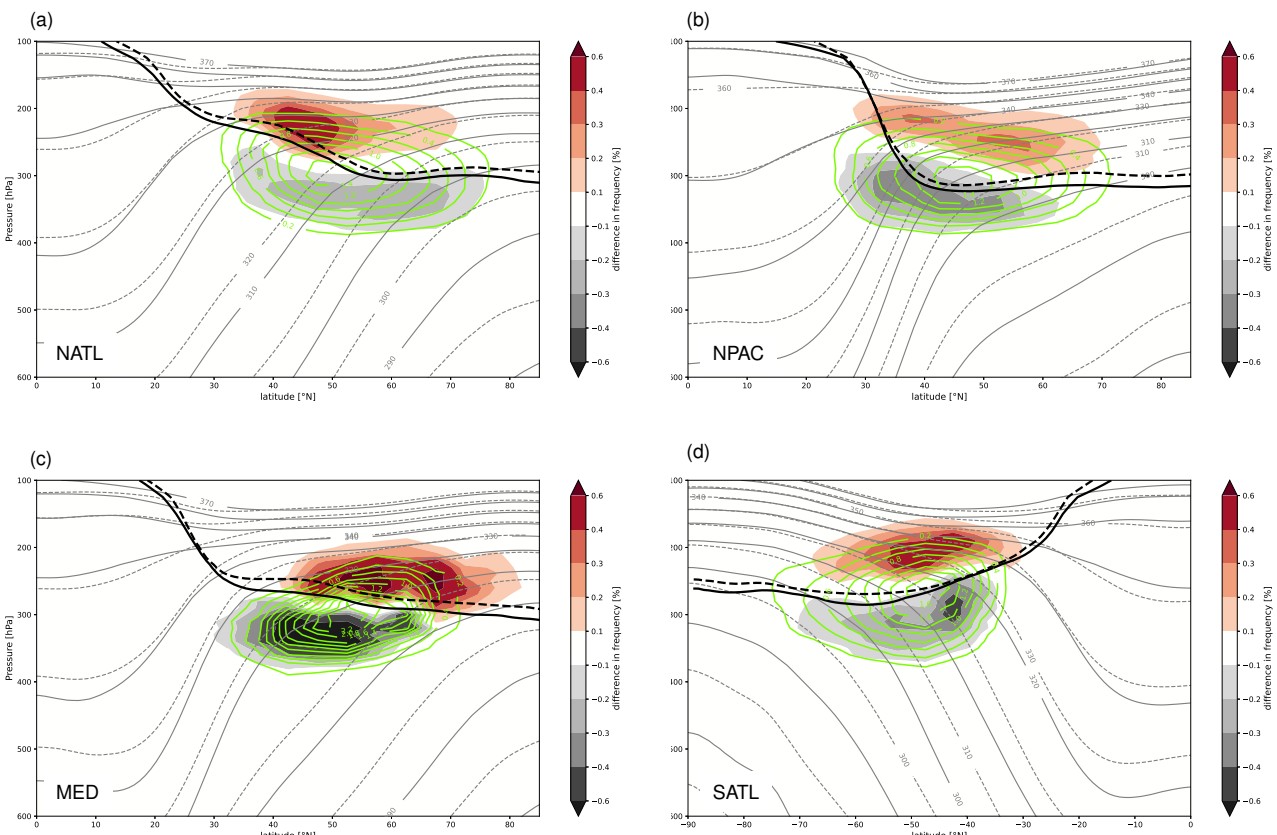

**Figure 11.** 50-year mean potential temperature (solid/dashed grey lines) and potential vorticity (2 pvu isolines; solid/dashed thick black lines) zonally averaged over the four target regions NATL, NPAC, SATL and MED. Dashed lines correspond to RCP85, solid lines to HIST. Green solid lines show the distribution of the WCB air parcels at the end of their ascent for HIST. The corresponding shift in a future climate, RCP85 minus HIST, is shown in color shading. All panels are for winter in the corresponding hemisphere, i.e., DJF in (a, b, c) and JJA in (d).

shift in the WCB outflows (see colour shading). Given that the dynamical tropopause remains rather unchanged in RCP85, this points to WCBs being able to more strongly disturb the upper-level flow, with implications for the downstream weather evolution and predictability, as discussed before. This finding is, of course, also consistent with the isentropic level reached at the end of the WCB ascent (see Figure 8d). For instance, in the NATL the end-of-ascent isentropic level changes (in the median) from $\sim 313\,\mathrm{K}$ in HIST to $\sim 322\,\mathrm{K}$ in RCP85, which fairly well matches with the isentropic levels in Figure 11 and supports the finding that the outflow is shifted closer to the tropopause.



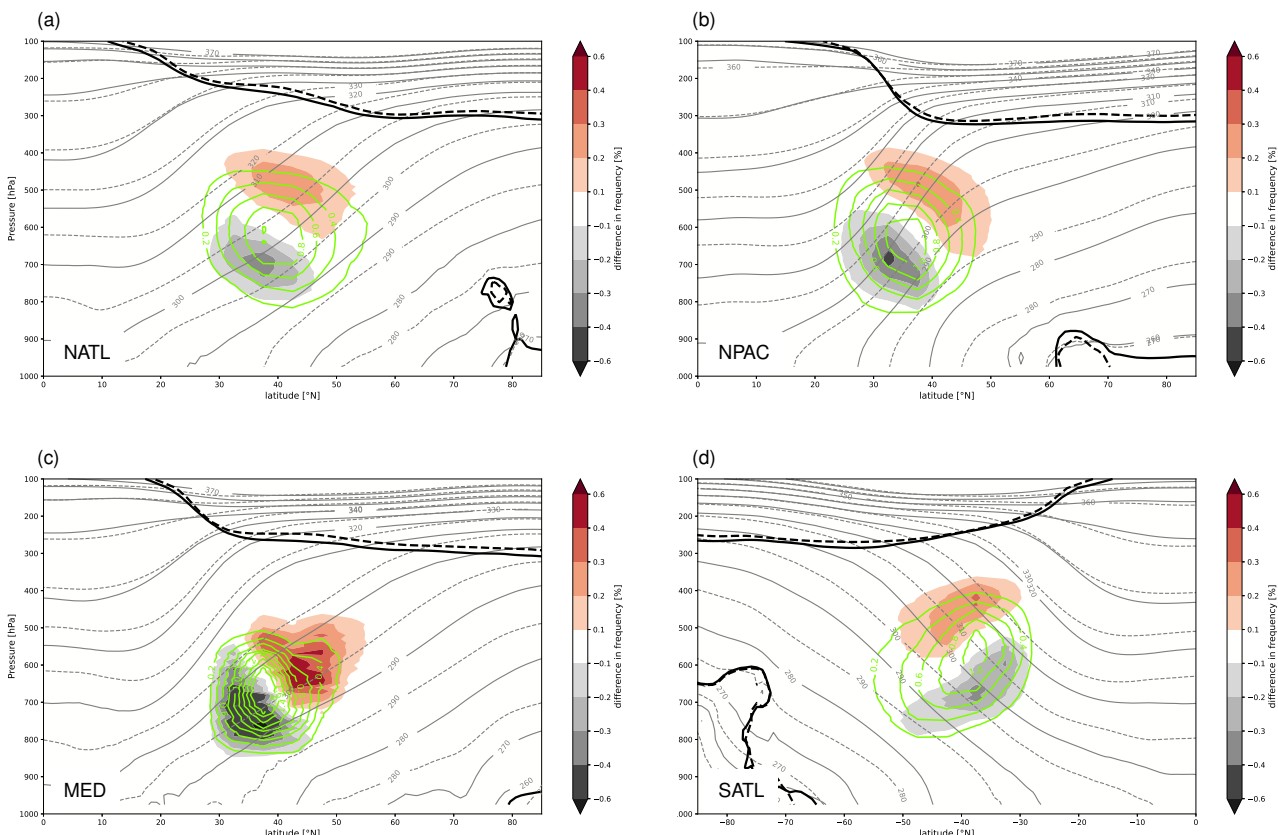

**Figure 12.** 50-year mean potential temperature (thin grey solid and dashed lines), potential vorticity (2 pvu isolines; solid and dashed bold black lines), zonally averaged over the four target regions NATL, NPAC, SATL and MED. Dashed lines correspond to RCP85, solid lines to HIST. Green solid lines show the distribution of the WCB air parcels during the time of maximum diabatic heating for HIST. The corresponding shift in a future climate, RCP85 minus HIST, is shown in color shading.

## 5.2 Mid-tropospheric diabatic heating in WCBs

In Figure 12, a frequency distribution of the location (altitude vs. latitude) of the maximum diabatic heating rate (DHR) occurring in the WCBs is shown as a zonal mean over the four regions. The maximum DHR in HIST is located at a height of 600 - 700 hPa in NATL, NPAC and MED, whereas it is located at slightly higher altitudes (500 - 600 hPa) in SATL (see green lines). The change in the position of the maximum DHR from HIST to RCP85 is shown in colour. In all considered regions, the maximum in the DHR shifts upward and poleward. Particularly in the MED, the latitude of maximum diabatic heating is considerably shifted ∼5° northwards. In contrast, the vertical shift of maximum diabatic heating is equally pronounced in all regions. The location of maximum heating rises to higher altitudes (lower pressure values). Additionally, the absolute values of the maximum DHR strongly increase by about 2-3 K $(6\,\mathrm{h})^{-1}$ from present-day to future climate, whereas the increase in the upper percentiles is slightly more pronounced with 3-4 K $(6\,\mathrm{h})^{-1}$ (see Supplementary Material, Figure S3). This signal is





consistent with the increase in precipitation and $\Delta\theta$ along the ascending WCBs, as shown in Figure 8b and c, and supports the increased diabatic nature of WCBs in a future climate. The increase in the maximum DHR and the shift to higher altitudes potentially also have an effect on the intensification of the associated cyclones, caused by changes in the associated diabatic modification of potential vorticity, as discussed in more detail by Binder et al. (2022). On the other hand, changes in the location

and amplitude of diabatic heating in the extratropical storm tracks can have an impact on the location of the eddy-driven jet (e.g., Lachmy and Kaspi (2020)), whereas Papritz and Spengler (2015) discuss the importance of mid-tropospheric diabatic heating for the slope of isentropic surfaces as a measure for baroclinicity. They showed that in the upper troposphere, the diabatic heating due to cloud processes is the dominant mechanism maintaining the isentropic slope. The observed changes in the DHR from the present to future climate could thus potentially influence the jet location and baroclinicity. A detailed

analysis of these effects is however beyond the scope of this work.

In summer, the position of the maximum in the DHR is located slightly higher than in winter, however a very similar upward and poleward shift in the maximum DHR is observed as in winter. The absolute values of the DHR are higher compared to winter, whereby the changes in the upper percentiles are even more pronounced in the future climate (see Supplement Figure S4).

## 5.3   Geographical distribution of WCB-related precipitation

In Figure 8c, it has been shown that the WCB-related precipitation increases in a warmer climate whereas the most pronounced change is seen in the upper percentiles. In Figure 13 we therefore discuss in more detail the geographical distribution of WCB-related precipitation and the changes in a future climate. In Figure 13a, the 50-year climatology for total precipitation is shown for DJF. Precipitation maxima occur over the inter-tropical convergence zone as well as in the main storm track regions and

at high mountain ranges (e.g., Rocky Mountains, west coast of Norway). In the North Atlantic and Pacific, precipitation is highest in the centre of the storm tracks, whereas in the southern hemispheric storm tracks a poleward decrease in precipitation is observed with exceptions for areas with high mountain ranges. In Figure 13b, the difference in total precipitation (RCP85 minus HIST) is shown. In the North Pacific storm track, total precipitation increases in the northern part, whereas a decrease is discernible in the southern part. In the North Atlantic, precipitation increases in the southwestern and northeastern part of

the storm track and decreases to the south and west of Iceland as well as over the eastern Mediterranean. In the Southern Hemisphere, precipitation increases over most parts of the storm tracks, with exceptions to the east of Madagascar and in the central South Pacific. The changes in total precipitation shown here for the RCP85 simulation are very similar to changes observed in other CESM1 simulations (e.g. Meehl et al., 2013) or in CMIP5 multi-model mean precipitation (e.g. Knutti and Sedlácek, 2012; Giorgi et al., 2019).

Figure 13c shows the percentage of total precipitation linked to WCBs. In the storm track regions, where also the frequency of WCB occurrence is highest (see Figure 4a,b), WCBs are responsible for up to 50% of precipitation and in the North East Atlantic, close to Iceland, they are still responsible for ~25%. In the Southern Hemisphere more than 50% of the total precipitation is associated with WCBs downstream of South America, Africa and Australia. This result is in very good agreement with the one presented in Pfahl et al. (2014). There, it was investigated how much precipitation can be attributed to WCBs in





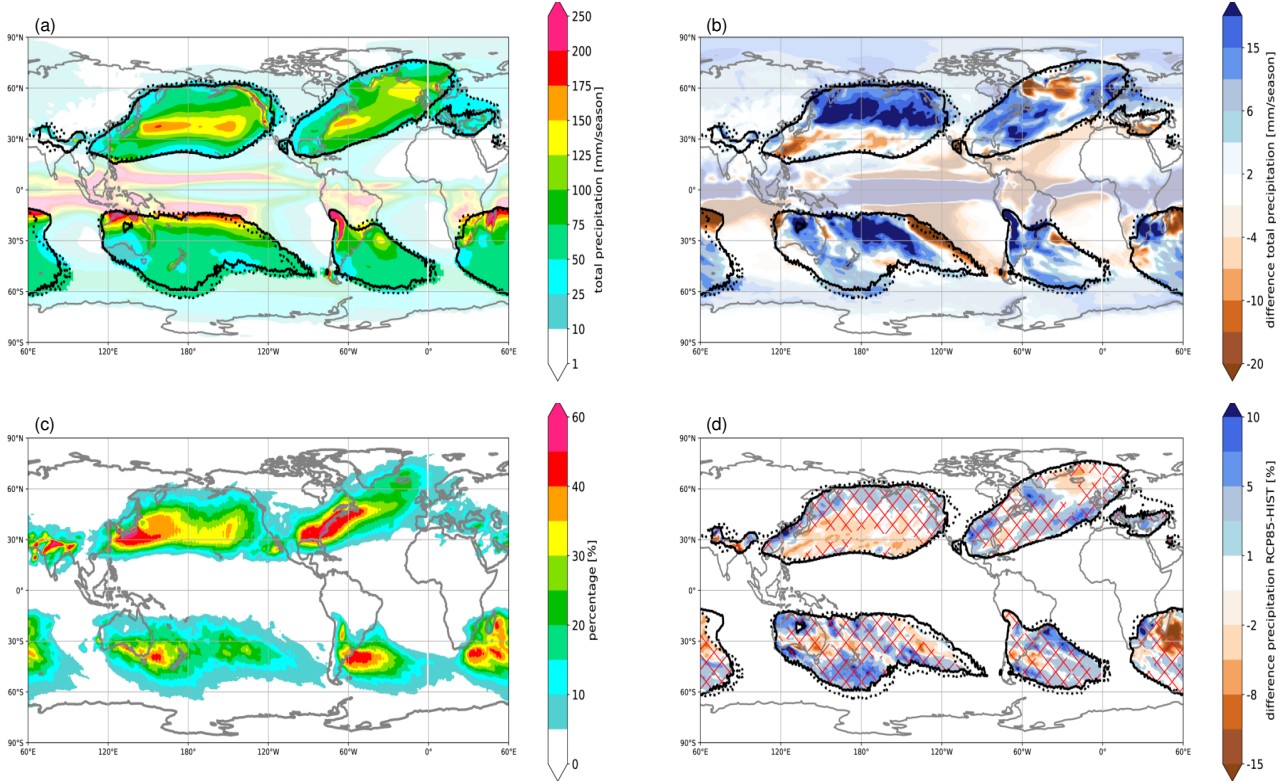

**Figure 13.** (a) 50-year climatology of total DJF precipitation ( mm /3 months$^{-1}$) for HIST, (b) difference in total precipitation between RCP85 and HIST, (c) percentage of precipitation that is linked to WCBs in DJF, and (d) difference RCP85 minus HIST for percentage of DJF precipitation linked to WCBs. Black solid and dashed lines denote the 1% occurrence frequencies for grid points that are part of a WCB during its inflow and ascent phase, thus while the air parcels are located below 400 hPa. Red hatched regions show where there is an increase in overall precipitation, thus they correspond to the blue colours in panel b). Fields are shown in transparent in all regions where the occurrence frequencies of WCBs in the inflow and ascent phase is less than 1%.

the ERA-Interim data set and it was found that in DJF more than 60% of total precipitation is associated with WCBs in the main storm track regions (see their Figure 7b). These results also confirm that the CESM model is able to reasonably capture the dynamical and microphysical processes that lead to precipitation.

The difference in the percentage of WCB-related precipitation between RCP85 and HIST is shown in Figure 13d, but only in
5    regions where the occurrence frequencies of WCBs in their inflow and ascent areas exceed 1% (black solid and dashed lines). The red hatched areas highlight regions where there is an increase in total precipitation in a future climate (blue colours Figure 13b). Changes in the percentage of WCB-related precipitation can occur because in a certain region the frequency of WCBs changes or because a WCB can become more or less effective in the formation of precipitation. Overall, the changes in WCB-related precipitation are small. In some regions, like in the North Pacific, at the U.S. East Coast, or on the polar side of the
10    southern hemispheric storm track, an increase in total precipitation together with an increase in the WCB-related precipitation





is observed. As mentioned above, this signal can either be caused by the increased inflow moisture of WCBs, by an increased occurrence frequency of WCBs or by an interplay of both effects. However, in these regions the potential for WCB-related extreme precipitation might be increased in a warmer climate. In other regions, e.g., close to Iceland or Madagascar, a decrease in WCB-related precipitation might be caused by a decrease in the occurrence frequencies. The most important changes in

WCB-related precipitation are, however, already shown in Figures 8c, 9c and 10, where it can be seen that the overall amount of precipitation falling along a WCB increases in a future climate, especially in the upper percentiles. This points to a potential increase in extreme precipitation events linked to WCBs in a warmer world.

## 6    Discussion and conclusion

In this study we made use of 50 years of present-day (1990-1999) and future (RCP8.5 scenario; 2091-2100) climate simulations

(CESM1) at six-hourly temporal and $\sim 1°$ spatial resolution to assess if WCBs can reasonably be represented in climate models, and to determine if their frequency of occurrence, geographical distribution and characteristics change in the warmer climate. The WCBs are identified based on a Lagrangian diagnostic: (i) kinematic air mass trajectories are released equidistantly (80 km mesh size) and at a six-hourly time interval in the near-surface layer (up to 700 hPa) over the whole globe; (ii) an ascent criterion of 600 hPa within 48 h is then applied to capture the WCB ascent, and the neighbourhood of the air parcel ascent to

an extratropical cyclone is required to identify them as one of the characteristic airstreams of an extratropical cyclone. The resulting 50-year climatologies of WCBs in HIST and RCP85 build the basis of this study. They are also compared to an existing 37 year climatology of WCBs based on ERA-Interim (1980-2018). Based on these data sets, and closing the circle to the research questions raised in the introduction, the main results of the study can be summarized as follows:

  – The climate simulation HIST produces frequency maps of WCB occurrence that capture many hotspots that have pre-
viously been identified in reanalysis data sets (e.g., ERA-Interim). In particular, WCB hotspots do occur in the North
       Atlantic and North Pacific storm track regions, and similarly in the storm tracks of the Southern Oceans. Thereby, also
       the seasonal cycle and the frequency amplitudes are comparable to ERA-Interim. Secondary WCB peaks are also cap-
       tured, e.g., in the Mediterranean. Finally, also the globally averaged ascent behaviour (time-pressure evolution) of the
       HIST WCBs is remarkably similar to the ERA-Interim one. Still, local discrepancies – of course – exist between the
climate simulations and the reanalysis data. This, however, must be expected due to natural variability, and because all
       of the discrepancies look physically plausible, we assume that CESM is able to realistically represent WCBs.

  – In a future (RCP85) climate the main WCB frequency hotspots remain similar to the ones for the present-day (HIST)
       climate, both in winter and in summer. Still, some geographical shifts in the frequency patterns are discernible, in
       addition to an overall increased amplitude (number of WCB trajectories) in RCP85. For instance, the WCB inflow
regions in boreal winter (DJF) are systematically shifted towards the north in the western North Pacific. In the North
       Atlantic storm track region, the shift takes rather the form of a southwest-northeast dipole, with increased frequencies
       in the southwestern North Atlantic and a decrease in the northeast, to the south of Iceland. An increase is also seen in





the South Atlantic, to the east of South America's coast, and a decrease near Madagascar. Of course, all these signals spatially spread out considerably during the WCB ascent and particularly until their outflow reaches near-tropospheric levels. In boreal summer (JJA), the shifts in the Northern Hemisphere are weaker, and an additional signal occurs over the North American continent; in the Southern Hemisphere, the South American shift remains, and actually confines to the

west a band of enhanced WCB inflow frequencies spanning to the east up to Australia. The climate-change shift in WCB frequencies regionally and partly, but not consistently, go along with corresponding changes in cyclone frequencies. This strongly indicates that the WCB changes cannot solely be attributed to cyclone frequency changes, but that other dynamical and thermodynamical factors determining the WCB-efficiency of extratropical cyclones must also be taken into account.

– The globally averaged ascent behaviour of all WCBs remains rather similar in RCP85 compared to HIST, but significant differences in WCB characteristics emerge if regional WCB hotspots (North Atlantic, North Pacific, Mediterranean, South Atlantic) are considered. The inflow moisture for all these regions is considerably enhanced, which consistently also leads to enhanced 48-hour accumulated precipitation during the WCB ascent. The latent heating due to precipitation, in turn, is associated with correspondingly enhanced diabatic heating rates (changes in potential temperature) in the mid

troposphere, and – consistently – higher potential temperatures at the end of the WCB ascent, as the air parcels reach the upper troposphere / lower stratosphere. These climate-change-related effects are discernible in winter, and (with partly larger amplitudes) also in summer. A statistically more refined analysis of the changes in the accumulated precipitation points to more extreme values. More specifically, particularly the most extreme percentiles of the precipitation increase substantially in RCP85 compared to HIST, and this consistently in all considered regions. A geographical analysis of

changes in WCB-related precipitation shows that the percentage of precipitation associated with WCBs increases in large parts of the storm tracks where also total precipitation increases in a future climate, pointing towards an increased efficiency of WCBs in precipitation formation. However also regions are observed where the percentage of WCB-related precipitation is decreasing despite an increase in total precipitation. Whether the increase in WCB-related precipitation is caused an increased occurrence frequency of WCBs or by the higher moister content or both cannot be disentangled

in this study.

  – the more diabatic nature of WCBs in a future climate leads to two important aspects: (i) the WCB outflow in a future climate is shifted upwards and closer to the tropopause in all considered regions. WCBs therefore have an increased potential to disturb the upper-level waveguide and the downstream flow evolution. (ii) the maximum in the diabatic heating rate is shifted slightly poleward and upward and increases in amplitude which can have an impact on the restoration of

baroclinicity and influence the jet location.

This study puts the focus on a weather-system-based perspective on climate change, i.e., WCBs were identified in six-hourly time steps as a specific synoptic-scale weather system and then, based on these distinct features, climatological aspects (WCB frequencies, characteristics and impacts) were discussed. Such a system-based approach, in comparison to more field-based approaches (e.g., by considering geopotential variability), has already successfully been applied for other weather systems,



most prominently for extratropical cyclones and atmospheric blocks.

In several studies, the response of extratropical cyclones to climate change as well as the representation of cyclone structure in climate models has been investigated based on a weather system perspective. To this aim, extratropical cyclones have been identified and tracked in global climate models or aquaplanet simulations and their structure has been evaluated. While the

overall structure of extratropical cyclones is represented well in climate models (Catto et al., 2019), problems still exist, e.g., in: the representation of precipitation (Hawcroft et al., 2016) and diabatic processes (Hawcroft et al., 2017), the link between cyclone dynamics and clouds (Govekar et al., 2014), and the relative humidity distribution in the WCB which is linked to the isentropic ascent (Catto et al., 2010). Future climate simulations predict changes in the wind speed in the warm sector of cyclones (Priestley and Catto, 2022), whereas Dolores-Tesillos et al. (2022) link these changes to enhanced diabatic heating,

which in turn leads to an amplified low-level potential vorticity (PV) anomaly and a change in the upper-level dipole PV anomaly. This is consistent with findings by Binder et al. (2022) who, in a future climate, also detects a stronger diabatic PV production in WCBs that in turn lead to enhanced cyclone deepening rates and a stronger intensity of the strongest cyclones. For a detailed description of the response of extratropical cyclones to climate change, also from a weather-system perspective, the reader is referred to Catto et al. (2019).

Another important weather system in the extratropics are atmospheric blocks, because they can be linked to extreme weather like cold spells (e.g., Sillmann et al., 2011; Bieli et al., 2015), heat waves (Quandt et al., 2019), and heavy precipitation events (e.g., Martius et al., 2013; Kautz et al., 2022). However, so far, the confidence in projected changes in atmospheric blocking is low as different physical mechanisms determine the whole blocking life cycle (Woollings et al., 2018). As WCB outflows are associated with low PV values and can therefore modify the upper-level flow, they can play an important role in the initiation

and/or maintenance of atmospheric blocks (Steinfeld and Pfahl, 2019; Steinfeld et al., 2020; Pfahl et al., 2015a). As WCBs become more diabatic in a future climate (see Figure 8c) and their ascent is located at higher altitudes/isentropes in future, it would be interesting to investigate in more detail the impact of WCBs on blockings in a future climate. The weather system perspective presented here would be an option to gain more insight into the underlying physical processes.

Furthermore, other weather features like the jet stream, Rossby waves or Rossby wave breaking events are important drivers

of the surface weather in the mid-latitudes. Extending the weather system perspective and its connection to WBCs in present-day and future climate simulations could, therefore, also lead to an improved understanding of the underlying physical mechanisms and their change in a warmer climate.

The current study comes with some caveats, the most restrictive one related to the small sample size (50 years) of the CESM simulations. This does not allow for a robust statistical analysis of the differences in WCB occurrence between the present-day

and the future climate, and – similarly – the corresponding comparison between the present-day climate and ERA-Interim data. For instance, we cannot ultimately determine whether a regional difference between ERA-Interim and HIST reflects a systematic bias between the two models, or whether it only reflects natural variability. A robust statistical analysis would proceed as follows: about 1'000 years (seasons) of CESM simulations would be performed and this would result in correspondingly many maps of WCB frequencies, thus representatively covering the full spectrum of WCB maps in the CESM climate. ERA-Interim,

as one realization of nature, would then have to be placed in comparison to this CESM distribution. If the ERA-Interim real-

ization, at a specific region, falls well within the CESM distribution, they can be assumed to be physically consistent. On the other hand, if the ERA-Interim realization lies at the extreme borders of the CESM distribution, we would have to assume that CESM exhibits a systematic bias in the WCBs frequencies due to deficiencies in capturing the essential WCB-related physics. An analogue statistical analysis would be required to assess the significance of differences between present-day and future

WCB maps. Such a refined statistical analysis would be very welcome, but it is computationally still prohibitive because of the high cost of trajectory calculations. As an alternative and to overcome this limitation, one could try to identify WCBs in climate models based on their Eulerian fingerprints (Quinting and Grams, 2021; Wandel et al., 2021).This, however, comes with its own caveats, e.g., by not allowing the WCB characteristics (latent heating) to be assessed during the WCB air parcels' ascent. Despite this limitation of the study, we think that all patterns found are physically plausible and thus should be taken as

reasonable first estimates how climate change impacts on WCBs.

In summary, the current study points to potential changes in WCB frequencies, intensities and characteristics in a warmer climate, but also some WCB impacts (precipitation, diabatic heating, WCB-jet interaction) have been shown to regionally change. In a forthcoming study, we intend to extend this WCB-impact aspect on an even broader and more systematic perspective, by following the methodology developed in Joos (2019). In particular, it will be rewarding to see how WCBs influence the surface

energy balance and (heavy) precipitation in the extratropics, the Arctic and Antarctica. We hope that this weather-feature-based approach to climate impacts will become a fruitful and important contribution to climate science.

*Code and data availability.* The WCB and cyclone data for CESM and ERA-Interim are available from the authors upon request.

*Author contributions.* UB provided the CESM simulations, and MS calculated the basic WCB trajectory climatologies based on CESM. HJ, HB and MS did the global and regional WCB analysis of the study. MS, HJ and HW conceptually developed the study. All authors

contributed to the writing.

*Competing interests.* The authors declare that they have no conflict of interest.

*Acknowledgements.* We thank Meteoswiss for providing access to the ERA-Interim data set. We are grateful to Matthias Roethlisberger for help with the CESM data and for fruitful discussions on the statistical significance. HB received funding from the Swiss National Science

Foundation (project 185049) and from the European Research Council H2020 research and innovation program (INTEXseas, grant no. 787652).



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
