# Peer review of "Warm conveyor belts in present-day and future climate simulations. Part I: Climatology and impacts"

_Weather and Climate Dynamics, 2022_

## Author Comment (AC1)

**Reply to**

**Comment on wcd-2022-38**
Anonymous Referee #1
* * *
Referee comment on "Warm conveyor belts in present-day and future climate simulations. Part I: Climatology and impacts" by Hanna Joos et al., Weather Clim. Dynam. Discuss., https://doi.org/10.5194/wcd-2022-38-RC1, 2022
* * *
**Synopsis:**
The present study by Joos et al. is the first to investigate changes of warm conveyor belts in a future climate. To do so, the authors have identified WCBs via trajectory calculations in 50 years of CESM simulations. Key findings are that CESM simulations reasonably well represent WCBs in present climate, that changes in WCB occurrence frequencies largely coincide with changes in midlatitude storm tracks, and that WCB related precipitation will increase due to an overall higher moisture content. Further, a poleward and upward shift of diabatic heating will allow WCBs to more favorably interact with the upper-level Rossby waveguide. The study is well written, the methods are established and the results are certainly of interest to the readership of WCD. I therefore recommend to accept this paper pending on some minor revisions.

Many thanks for this positive feedback. We are happy that the reviewer thinks that the results will be of interest to the *Weather and Climate Dynamics* readership and that it is well written. Still, we appreciate very much the minor comments by the reviewer and will address them below point by point. They will definitely help to improve the clarity of the manuscript.

**Minor:**

p.1, l.2: What is meant by "strong cloud formation"? Perhaps simply remove "strong"?

We will remove 'strong', as suggested by the reviewer. When writing this line, we had in mind that WCBs are the main precipitation- and cloud-forming weather systems in the extratropics, where the clouds can be either of large-scale and/or convective nature. 'Strong' was meant to indicate the relevance of WCBs for cloud formation in the extratropics. But we agree with the reviewer that the word 'strong' is not necessary.

p.1, l.6: Should it be 1990-1999 instead of 1991-2000?

Yes, we will correct it in the abstract.

p.1., l.13: Could you be more specific how the characteristics change, e.g., increase or decrease?

We will be more specific in the revised manuscript. The changes mentioned in the abstract refer to the ones seen in Figure 8 and 9. The values generally increase in a future climate,

whereby the spatial distribution for precipitation also exhibits some interesting regional aspects.

p.1, l.16/17: Does "strong" mean that the increases are significant? Perhaps choose a more quantitative description by providing the relative or absolute increase.

The findings we are referring to in the abstract are based on the box-whisker plots in Figures 8 and 9. We have not applied a formal statistical test to see whether the changes in total precipitation and diabatic heating (potential temperature) between present-day and future climate are statistically significant. In this sense, we clearly do not mean 'strong' as 'statistically significant', and we will change the wording in the abstract accordingly.
Still the changes in Figures 8 and 9 are substantial – a word that we might use instead of 'strong'. For instance, if we consider the change of diabatic heating for the region NATL in Figure 8b, the values change from ~20 K in present-day climate to ~25 K in future climate. Hence, the percentage change amounts to 25%. To make the statement in the abstract more quantitative, we will give a range of percentage changes for total precipitation and diabatic heating. This range will then reflect the changes for the different regions.

p.2, l.5: The authors may want to reconsider the structure of the sentence. Better write "...deep convective clouds associated with the Hadley-Walker circulation dominate in the tropics...."?

Thanks, we will change the sentence according to the suggestion.

p.2, l.24: WCBs do not necessarily ascend poleward. Perhaps adding "most often"?

Perfectly right. Thanks for this imprecise wording. We will include 'most often' in the text.

p. 4, l.5: To better understand possible differences between ERAI and CESM the reader may need some more information concerning the CESM setup. Could you therefore include information on the native grid spacing of the CESM simulation and how this compares to ERAI? Also, it could be worthwhile to mention that the number of vertical levels is considerably higher in ERAI than in CESM.

Many thanks for pointing us to these missing pieces of information. We extend section 2.1 in this sense. ERA-Interim has a grid resolution of approximately 80 km, whereas the resolution is, with 100 km, somewhat coarser in CESM. More important, however, is the difference in the number of vertical levels. Whereas there are 60 levels in ERA-Interim, there are only 30 levels in CESM.

p. 4, l.12: I understand that the identification of WCBs in CESM is nearly identical to that in ERAI. Still, could you clearly mention possible differences (e.g., number of vertical levels).

In both, ERA-Interim and CESM, the starting grid for the WCB trajectories is exactly the same, i.e., the air parcels are released on a horizontally equidistant grid with 80 km distance, and vertically they are released from 14 equally spaced pressure levels between 1050 and 790 hPa. This piece of information is already included in section 2.2. However, we agree with the reviewer, that it is worthwhile to highlight more explicitly that CESM has only 30

model levels, compared to 60 levels in ERA-Interim, and that this difference might affect the quality of the trajectory calculation. We will mention this potential 'problem' in the revised manuscript, but will also anticipate already in section 2.2 that the WCB characteristics of the CESM trajectories match fairly well with the ones of ERA-Interim (see section 3), thus confirming CESM's capability to reasonably represent WCBs (trajectories and climatologies).

p. 5, l.14: If I recall correctly, ERAI WCB trajectories of Sprenger et al. (2017) were mapped to 1° grid spacing. What is the motivation for choosing 0.5° in CESM? Were the ERAI data remapped to the same grid spacing prior to comparing the two data sets. If not, could the different resolution explain some of the differences seen in Fig. 2?

This is an important point. Many thanks for pointing us to this unclarity in the original manuscript. Of course, the ERA-Interim WCB climatology was re-mapped to the 0.5 x 0.5 latitude/longitude grid used in this study. More specifically, we took the previous WCB trajectories of the ERA-Interim climatology in Sprenger et al. (2017), which matches with Madonna et al. (2014), and treated them exactly in the same way as the new ones in the CESM WCB climatology. Thus, we can be sure that the gridding of the two data sets is perfectly equivalent, and that the differences in Figure 2 are not due to the gridding.
Why a 0.5 x 0.5 latitude/longitude grid, and not gridding CESM to the previous ERA-Interim 1 x 1 latitude/longitude grid? This is a valid question, and we considered this also in the beginning of the project. However, at a later stage, we would like to compare ERA-Interim and CESM also to a corresponding ERA5 WCB climatology, for which a higher spatial resolution is certainly appropriate. We refrained from including such a comparison in the study, because the horizontal (and vertical) resolution of CESM is much more comparable to ERA-Interim than to ERA5.

p. 6, l.2: I fully understand that the authors chose a period of 37 years from ERAI to have the largest possible sample size. Still I am wondering, if the authors calculated the ERAI WCB frequencies for the same period as with CESM (1990-1999) would the differences between the two datasets become even smaller?

This is an important point. There are, actually, different aspects to consider: First, by restricting the ERA-Interim calculation to the same period as in CESM, we would make the comparison period (10 years) of the two data sets equivalent, but would still have an imbalance of (then) ten years ERA-Interim vs. 50 years of CESM. We would, secondly, also restrict some of the external forcing included in the CESM simulation to correspond to the ERA-Interim forcing. This would apply, for instance, to the Pinatubo eruption in 1991. However, since CESM is a coupled atmosphere-ocean simulation, some important other forcing factors cannot be expected to match with observed forcings. For instance, the ENSO variability in CESM cannot be expected to match better with the observed variability.
In summary, even if we restrict the ERA-Interim period to 1990-1999, it is not immediately clear whether we can expect a better agreement between CESM and ERA-Interim, or not. On the other hand, extending the ERA-Interim period to 37 years, we expect to have a more robust estimation of the average WCB climatology, which also is more comparable to the 50-year climatology of CESM. The challenge and the limitations in comparing CESM and ERA-Interim WCB climatologies is discussed in Section 2.3. We will include a short discussion in this Section on the restriction of ERA-Interim to 1990-1999. Finally – because it is easily done – we will follow the reviewer's suggestion and will calculate the 1990-1999

WCB climatology in ERA-Interim. The result, if adding some new insight, will be included in the Supplement Material.

p.6, l.28/29: I assume it should be Northwest Pacific instead of Northeast Pacific.

Yes, we will correct it in the revised manuscript.

p. 6, l.32: The similarity between ERAI and CESM is indeed striking. Still, it seems to me that some differences may exist which are currently not seen due to the choice of figures. For example, the DJF WCB frequencies seem to be generally higher in ERAI than in CESM (Figs. 2a,b) and differences at t=0h and t=48h are hardly visible since only the 1% isoline is shown. Have the authors considered to include difference maps between the two? I would strongly encourage the authors to either include such difference maps in the main paper or at least in the supplemental material. For example, it would be interesting to see whether biases in WCB frequency at t=48h correspond to biases in blocking frequency reported in previous studies (e.g., Woolings et al. 2018).

Many thanks for this suggestion. Of course, we also considered showing difference maps in the manuscript. However, we then decided not to do so, actually exactly based on the ground that the reviewer him-/herself writes in the comment: "The similarity between ERAI and CESM is indeed striking." It is this similarity that we would like to highlight in the manuscript, i.e., to show that CESM does an excitingly good job in representing key features (climatological patterns, ascent behaviour) of WCBs. In this sense, we would prefer not to show difference plots in the main text.
However, we also fully understand the reviewer's point. It would indeed also be interesting to see in greater detail how the two climatologies compare. We also like the reviewer's suggestion to relate the biases in WCB frequencies at t = 48 h to corresponding biases in blocking frequencies. We will add a reference to Woolings et al. (2018) to make the reader aware of such (potential) links. Furthermore, as suggested by the reviewer, we will add the difference plots to the Supplement Material and will refer to them in the main text.
Finally, a word of caution. The patterns emerging from a difference plot must be carefully interpreted. We only have 37 years of ERA-Interim and 50 years of CESM-HIST data available, and it remains very challenging (or even impossible; see discussion in Section 2.3) to assess the statistical significance of the emerging patterns based on these relatively few years. Hence, part of the emerging patterns might 'only' reflect interannual variability.

p. 8, l. 1: What exactly is shown in the intensity maps of the supplementary material? I would understand if the unit of the intensity maps was "number of trajectories/6h". Currently it is in %. Is this correct? If correct please explain how to understand the intensity in %.

Many thanks for pointing us to this unclarity. Indeed, we have to apologize for the unclear and, actually, misleading unit. The interesting comparison is, e.g., between Figure 4 in the main text and the corresponding Figure S1 in the Supplement Material. We describe now in detail in which way the two figures differ.
Figure 4c, for instance, considers how often a CESM grid point is located nearby a passing WCB trajectory, where 'nearby' means that the trajectory has to pass by within a 100 km distance and 'passing' refers to the time instance 24 h since the start of the WCB trajectory.

The value is given as a percentage relative to all considered time instances. As an example: in the North West Pacific a WCB is passing by in 3-5% of all time instances.

Because Figure 4 provides 'only' frequencies, the information on how many WCB trajectories pass by is lost. If two WCB trajectories pass by within a 100 km distance, in the frequency map of Figure 4c it will only be considered that WCB trajectories passed by, but not how many did so. In contrast, as the reviewer correctly remarks, Figure S1c of the Supplement Material provides the information on how many trajectories are passing by. Hence, Figure 4 shows frequencies , and Figure S1 shows absolute numbers.

The correct unit of Figure S1 and S2 is '#trajectories / 6 h', what we refer to as a WCB intensity in the text. We will correct it in the revised manuscript.

p. 9, l.5/6: I guess it is 1990-1999 instead of 1990-2000.

Yes, we will correct it as suggested.

p. 9, l.23: The decrease of WCB frequency at t=0,24, and 48h east of Madagascar is a rather stationary signal, i.e., the decrease in WCB outflow does not occur far downstream of WCB inflow as one would expect in midlatitudes. So, are the signals seen here really WCBs or is it rather the signature of recurving tropical cyclones that are expected to become less frequent in a warmer climate (Fig. 3 in Roberts et al. 2020)? This decrease of tropical cyclone frequency likely corresponds to the decrease in cyclone frequency seen in Fig. 7 of the present manuscript. Since the authors already tracked cyclones in CESM: would it be possible to determine the genesis region of cyclones around Madagascar? In my view, this would aid the interpretation of the results. Depending on the outcome of this analysis please consider to change the abstract and conclusion accordingly.

Many thanks for this valuable comment. Indeed, we fully agree with the reviewer that the 'WCB signal' around Madagascar asks for a more careful discussion. We will definitely extend the discussion in 4.1 as suggested by the reviewer, i.e., by pointing out that the signal might actually be related to recurving tropical cyclones. Thanks also for the reference to Roberts et al. (2020), which we will add to the discussion in the text.

The reviewer is also right in that it would be rather easy to deepen the discussion by exploiting the cyclone dataset we have at hand. Hence, we gratefully will determine, as suggested, the genesis region of the Madagascar cyclones, and – depending on the outcome of this analysis – we will adapt the discussion, abstract and conclusions. We apologize that we are not yet able to provide a first, rough analysis in this direction in this reply document. However, as mentioned before, it will be easy to do and it will certainly be elucidating and help interpreting the results.

p. 9, l.32: To my understanding of Fig. 4f WCB outflow is less often located around 45°N rather than 30°N.

Thanks! Yes, of course the outflow is less often located around 45°N. We will correct it in the revised manuscript.

p. 10, l.1: The increase mostly occurs north of 60°N. Please correct.

Thanks, we will change it.

p. 10, l.15: Also here, please double-check the units of Figures S1 and S2.

See reply to the previous comment. We will make sure that the distinction between WCB frequencies (in the main text) and WCB intensities (in the Supplement Material) becomes clear in the revised manuscript.

p. 10, l.33: Please double-check the latitude. I would rather say that the increase is found around 60°N.

Correct. We will change it in the revised manuscript.

p. 11, l.1: As mentioned above: The decrease of cyclone frequency around Madagascar could be a signal of changed tropical cyclone frequencies. Perhaps the authors can refer to the corresponding literature (e.g., Roberts et al. 2018) at this occasion.

Many thanks. We will include this reference to the text, and – as mentioned before – will address/discuss the issue of the Madagascar cyclones and their "WCBs" more carefully in the revised manuscript.

p. 13, footnote: "WCB air parcels no longer increase in altitude, i.e., the pressure increases again" might be slightly confusing. Better write "...no longer gain altitude..."?

Thanks, we will change it.

p. 14, l.2: Is the increase in specific humidity what one would expect based on the Clausius-Clapeyron relation?

This is a good point. We consider, for instance, in Figure 8 the signals for the region NATL. The median of specific humidity increases from ~9.5 g/kg in present-day climate to ~12 g/kg in future climate. To relate this change in specific humidity to the Clausius-Clapeyron relation, the corresponding change in the temperature in the WCB inflow is needed. We will extract this piece of information from the WCB data sets in the two CESM climate simulations and will then be able to see if the relation holds or not.
As a reference, we already cite here the Clausius-Clapeyron relationship according to the Glossary of Meteorology of the American Meteorological Society (glossary.ametsoc.org/wiki/Clausius-clapeyron_equation):

$$e_s(T) = 0.6112 \, \exp\left( \frac{17.67\,T}{T + 243.5} \right),$$

Here, T is temperature in °C and $e_s(T)$ is the saturation vapor pressure in kPa. This formula is valid in the temperature range between -35°C < T < +35°C according to Bolton (1985). Using this approximate formula, one can show that at terrestrial conditions (T~260 K) a temperature change of 3 K results in a 20% change of the saturation vapor pressure. Assuming that the relative humidity remains fixed, the actual water vapor accordingly also

increases by 20%. In summary, as a first approximation a 20% increase in specific humidity is expected for every 3 K temperature increase.
We will check whether this really applies for the WCB inflows, and will accordingly discuss it in the revised manuscript.

p. 14, l.8: I assume that the distinction between large-scale and convective precipitation is based on the classification by CESM (parametrized vs resolved). Please clarify in the text.

Yes, exactly. There is, of course, no other way to distinguish between convective and large-scale precipitation in the climate model. We will write explicitly, as suggested by the reviewer, that the classification is based on the CESM classification.

p. 17, l.6: I guess it is 5.0mm/48h instead of 50mm/48h.

Yes, we will correct this to 5 mm / 48 h in the revised manuscript.

p. 18, l.31: I understand that the maximum anomaly is located within the lower stratosphere. But is this also true when considering the absolute frequencies? Further, I am wondering would the increase in outflow frequency mean that transport across the (dynamic) tropopause increases? Have you considered to quantify the fraction of WCBs that eventually end up in the lower stratosphere based on their PV at t=48h. An increase of WCB outflow in the lower stratosphere would raise interesting questions concerning their contribution to stratospheric water vapour concentrations in a warming climate - a discussion which focuses often on deep convection (e.g., Smith et al. 2022).

This is a very interesting point, which we missed to address in the original manuscript. Many thanks for making us aware of it. The moistening (or the dryness) of the stratosphere is, of course, a research topic of high relevance, and – as the reviewer comments – the water vapour input is often related to (over-shooting) convection. Our results, however, indicate that more water vapour could be transported across the mid-latitude tropopause by means of the large-scale WCBs. We will follow the reviewer's suggestion by: (i) discussing in Section 5.1 the effect of WCBs on stratospheric moistening and its potential increase in a future climate in the manuscript; (ii) calculating the fraction of WCB trajectories that reach the stratosphere for the present-day and future climate. The calculation in (ii) will then allow for a more quantitative assessment, in addition to the more speculative statements in (i).
We also very much like the suggestion by the reviewer because it allows us to refer to previous studies on stratosphere-troposphere exchange that the authors did in the past (e.g., Skerlak et al., 2014). Hence, we intend to do a first simple analysis on the impact of WCBs and their trends in this study, and take the reviewer's suggestion as a welcome 'invitation' to investigate this aspect in greater detail in forthcoming studies.

Škerlak, B., Sprenger, M., and Wernli, H.: A global climatology of stratosphere–troposphere exchange using the ERA-Interim data set from 1979 to 2011, Atmos. Chem. Phys., 14, 913–937, https://doi.org/10.5194/acp-14-913-2014, 2014.

p. 20, l.2: Is the diabatic heating rate calculated explicitly or is it based on the potential temperature change along the trajectories? Please explain.

Many thanks for having identified this unclarity. Indeed, we determine the change in potential temperature along the trajectories as the diabatic heating rate. More specifically, we consider along the trajectories all six-hour time intervals and then determine for these intervals the change in potential temperature.

As an alternative, explicit calculation of the heating rate, we would need 3D Eulerian heating rates as CESM model output, which are not available in the model output.

p. 20, l.6: I agree that the anomaly suggests a poleward shift by 5°, but is this also true when considering absolute values (i.e., the sum of HIST and anomalies)?

We are not sure whether we fully understand this point. Possibly, the wording is not perfectly clear. The color shading does not correspond to an *anomaly*, but is the difference (HIST-RCP85) between the distribution of maximum heating for the two climate simulations. Hence, the green contour lines give the distribution for HIST, and the contour 'lines' for RCP85 are obtained by adding the color-shaded field to the green-contour field. We assume that the reviewer interpreted the figure in this way, although we would not refer to the color shading as an anomaly.

Still, the reviewer raises an interesting question. Let's consider, for instance, Figure 12b, which shows the shift for the region NPAC. The frequencies for HIST (green lines) reach values up to ~1%, whereas the color shading is between -0.3% to +0.3%. Since the absolute and difference values are comparable in amplitude, we expect that the overall maximum is also shifted to the north in RCP85 compared to HIST. To make this point, however, we will add in the revised manuscript (Supplement Material) the plots corresponding to Figure 12, but with absolute values for HIST and RCP85.

If we see that the shift is not that clear in the new figures, we will discuss this shift to the north more carefully in the revised manuscript.

p. 21, l.30: Please include a brief description on how the percentage of total precipitation linked to WCBs is quantified (matching of WCB masks and precipitation fields I assume).

The description in the original manuscript is indeed too short and unclear. Many thanks for pointing us to this deficiency. The attribution of precipitation to the WCBs is done in exactly the way that the reviewer suggests. Hence, for all CESM grid points and time steps it is determined whether the total precipitation at that location and time coincides with a WCB mask. If it coincides, the precipitation is attributed to the WCB, if not we assume the precipitation to be independent of a WCB. The WCB masks used for this attribution are calculated as described in the final paragraph of Section 2.2. In particular note that each grid point at a specific time is labeled as a WCB point if a WCB trajectory passes by within a 100-km distance at the corresponding time.

A similar approach was applied in Joos (2019), where net top-of-atmosphere radiation was attributed to the presence and absence of WCBs. Furthermore, Pfahl et al. (2014) quantified in a similar way how the ERA-Interim precipitation relates to WCBs (or not).

Following the suggestion by the reviewer, we will add a few sentences in Section 5.3 that explain in greater detail the method.

p. 22, l.3: Is it really certain that the microphysical processes are reasonably captured in CESM? Perhaps this statement could be weakened by stating that the integrated effect of microphysical processes is reasonably captured.

Good point! Of course, the reviewer is perfectly right in pointing out that it is only the *integrated* effect of the microphysical processes that is captured reasonably well in CESM. His/her statement is more appropriate, and we will accordingly weaken the statement in the revised manuscript.

p. 24, l.28: To my understanding it is not only the changes in WCB outflow which have the potential to disturb the jet stream but also the changes in WCB ascent. A poleward shift of the ascent regions will lead to an irrotational outflow closer to the upper-level jet. If the authors agree this aspect could be included in the discussion.

This is an interesting point. We agree with the reviewer, and will accordingly adapt the text of the revised manuscript. However, the point raised by the reviewer is also a challenging one because it points to the ambiguity in defining the WCB outflow. Where should one set the boundary between the WCB ascent and the WCB outflow? In Figure 11 we adopted the rather extreme attitude that the outflow location is where the WCB trajectories reach their highest level (lowest pressure), before descending afterwards. One could argue that the interaction with the jet already occurs at lower altitudes, e.g., by invoking an irrotational flow field at upper levels that is able to interact with the jet. In previous studies, we also adopted the perspective that the WCB outflow shall be defined as all the levels where the WCB trajectories reach above 400 hPa. This would often identify more clearly the upper end of the WCB ascent, but is rather 'static' and underestimate the altitude of the WCB airstream.
In summary, we agree with the reviewer that the WCB airstream is associated at its upper tropospheric levels with an irrotational flow that potentially also disturbs the jet; but we also state that the location (altitude) of the WCB outflow is per se not well defined. We will address both aspects in the revised manuscript.

p. 25, l.6: Remove colon after "in".

We will do that.

p. 25, l.22: In a recent study Steinfeld et al. (2022) showed that the frequency of WCBs in blocking anticyclones is expected to increase by 15%. Perhaps the authors could refer to their study.

Many thanks for pointing us to this study. It nicely fits into our analysis and we will definitely refer to it in the revised manuscript. Actually, we had already in mind to do so, as one of the authors actually is a co-author of Steinfeld et al. (2022).

**Figures:**

Figs. 2, 4, 5, 7, 11, 12, 13: Please increase the font size of the axis labels.

Many thanks for these comments on how to improve the quality of the figures. We will increase the font size in all figures, as suggested. Furthermore, we decided that all geographical maps (Figs. 2, 4, 5, 7, 13) could/should be improved, e.g., by choosing a better geographical projection and by avoiding multiple colorbars whenever possible..

Figs. 11, 12: Please indicate that PV in the southern hemisphere has been multiplied by -1.

Many thanks. We will add this important piece of information.

**References:**

Roberts, M.J., Camp, J., Seddon, J., Vidale, P.L., Hodges, K., Vannière, B., Mecking, J., Haarsma, R., Bellucci, A., Scoccimarro, E., Caron, L.-P., Chauvin, F., Terray, L., Valcke, S., Moine, M.-P., Putrasahan, D., Roberts, C.D., Senan, R., Zarzycki, C., Ullrich, P., Yamada, Y., Mizuta, R., Kodama, C., Fu, D., Zhang, Q., Danabasoglu, G., Rosenbloom, N., Wang, H. and Wu, L. (2020), Projected Future Changes in Tropical Cyclones Using the CMIP6 HighResMIP Multimodel Ensemble. Geophys. Res. Lett., 47: e2020GL088662. https://doi.org/10.1029/2020GL088662

Smith, J. W., Bushell, A. C., Butchart, N., Haynes, P. H., & Maycock, A. C. (2022). The effect of convective injection of ice on stratospheric water vapor in a changing climate. Geophysical Research Letters, 49, e2021GL097386. https://doi.org/10.1029/2021GL097386

Steinfeld, D., Sprenger, M., Beyerle, U., & Pfahl, S. (2022). Response of moist and dry processes in atmospheric blocking to climate change. Environmental Research Letters, 17(8), 084020. https://doi.org/10.1088/1748-9326/ac81af

Woollings, T., Barriopedro, D., Methven, J. et al. Blocking and its Response to Climate Change. Curr Clim Change Rep 4, 287–300 (2018). https://doi.org/10.1007/s40641-018-0108-z

---

## Author Comment (AC2)

**Reply to**

**Comment on wcd-2022-38**
Anonymous Referee #2

Referee comment on "Warm conveyor belts in present-day and future climate simulations. Part I: Climatology and impacts" by Hanna Joos et al., Weather Clim. Dynam. Discuss., https://doi.org/10.5194/wcd-2022-38-RC2, 2022

**Synopsis:**
In this study the authors investigate how the global frequency, intensity, and characteristics of warm conveyor belts (WCBs) are represented in a state-of-the-art climate model, and how they change in a high emission warming scenario. The study is thorough, considering not only the spatial distribution of the WCBs, but also the changing characteristics and how these relate to the key impacts of WCBs - namely precipitation and large-scale dynamics. I find the paper to be very well written, clear, and easy to read. I have only a few minor comments/edits.

Many thanks for this positive feedback. We are happy that the reviewer thinks that the results will be of interest to the *Weather and Climate Dynamics* readership and that it is well written, clear and easy to read. Still, we appreciate the few minor comments by the reviewer very much and will address them below point by point.

**Minor:**
Page 3, Lines 13-14: I think it's not clear yet what "corresponding signals" might refer to. I understand having read the paper, but when I first saw this I was unclear. Could this be reworded?

'corresponding' is indeed one of the unnecessary words that found its way into the text. We will just omit it in the revised manuscript. When we wrote the text, of course, we already knew that the cyclone climatologies also exhibit some climate-change-related trends, and we already linked them to the WCB trends – thereby adding 'corresponding'.

Data: Could you say something about the different resolution of the datasets, and whether this has an impact on the results? For example, does the "intensity" of the WCBs, i.e., the number of trajectories in a grid box, depend strongly on the resolution?

This is a good point. Reviewer 1 also asked for further details on the model resolutions of CESM and ERA-Interim. We extend section 2.1 in this sense. ERA-Interim has a horizontal resolution of approximately 80 km, whereas the resolution is, with 100 km, somewhat coarser in the CESM simulations. More important, however, is the difference in the number of vertical levels. Whereas there are 60 levels in ERA-Interim, there are only 30 levels in CESM.
Hence, we do not expect the horizontal resolution to have a big impact on the trajectory calculation that is needed to identify WCBs. More critical is the reduced vertical resolution of CESM compared to ERA-Interim. The geographical patterns of the WCB climatology and

their ascent behaviour agree rather well between the two models. Hence, we are confident that the vertical resolution actually is not substantially affecting the WCB identification. Of course, a systematic analysis on the impact of the resolution on the calculation/identification of WCB trajectories would be interesting, but is beyond the scope of our study. To this aim, climate simulations (with one single model) at different resolutions would be necessary.

What we make sure in our analysis is that the starting grid and vertical levels of the trajectories are identical for CESM and ERA-Interim. Further, we also apply exactly the same postprocessing to the trajectories, i.e., with respect to gridding to the 0.5 x 0.5 latitude/longitude grid. In the revised manuscript we will highlight somewhat more that the CESM and ERA-Interim settings are identical, where possible.

Page 6, line 17: "chapter" should be "section".

Thanks. We will correct it as suggested.

---

## Author Response (AR1)

**Reply to**

**Comment on wcd-2022-38**
Anonymous Referee #1
* * *
Referee comment on "Warm conveyor belts in present-day and future climate simulations. Part I: Climatology and impacts" by Hanna Joos et al., Weather Clim. Dynam. Discuss., https://doi.org/10.5194/wcd-2022-38-RC1, 2022
* * *
**Synopsis:**

The present study by Joos et al. is the first to investigate changes of warm conveyor belts in a future climate. To do so, the authors have identified WCBs via trajectory calculations in 50 years of CESM simulations. Key findings are that CESM simulations reasonably well represent WCBs in present climate, that changes in WCB occurrence frequencies largely coincide with changes in midlatitude storm tracks, and that WCB related precipitation will increase due to an overall higher moisture content. Further, a poleward and upward shift of diabatic heating will allow WCBs to more favorably interact with the upper-level Rossby waveguide. The study is well written, the methods are established and the results are certainly of interest to the readership of WCD. I therefore recommend to accept this paper pending on some minor revisions.

Many thanks for this positive feedback. We are happy that the reviewer thinks that the results will be of interest to the *Weather and Climate Dynamics* readership and that it is well written. Still, we appreciate very much the minor comments by the reviewer and will address them below point by point. They will definitely help to improve the clarity of the manuscript.

**Minor:**

p.1, l.2: What is meant by "strong cloud formation"? Perhaps simply remove "strong"?

done

p.1, l.6: Should it be 1990-1999 instead of 1991-2000?

done

p.1., l.13: Could you be more specific how the characteristics change, e.g., increase or decrease?

We changed the corresponding sentence to:

Changes are also found in the WCB characteristics, as e.g., an increase in specific humidity of the WCB inflow, the WCB-related precipitation and the cross-isentropic ascent and the isentropic level reached by the WCB outflow

p.1, l.16/17: Does "strong" mean that the increases are significant? Perhaps choose a more quantitative description by providing the relative or absolute increase.

We included quantitative information about the increase and slightly reformulated these sentences to:

Changes are also found in the WCB characteristics, which has implications for WCB impacts in a future climate. The increase in inflow moisture in the different regions and seasons (~ 23-33% (~ 14-20%) in winter (summer) leads to: (i) an increase in WCB-related precipitation (~ 13-23% (~ 7-28%) in winter (summer)), especially in the upper percentiles, thus extreme precipitation related to WCBs might increase; (ii) a strong increase in diabatic heating (~ 20-27% (~17-33%) in winter (summer)) in the mid-troposphere; and (iii) a higher outflow level (~10 K (~ 10-16 K) in winter (summer)), which favours WCBs to more strongly interact with the upper-level Rossby waveguide.

p.2, l.5: The authors may want to reconsider the structure of the sentence. Better write "...deep convective clouds associated with the Hadley-Walker circulation dominate in the tropics...."?

done

p.2, l.24: WCBs do not necessarily ascend poleward. Perhaps adding "most often"?

done

p. 4, l.5: To better understand possible differences between ERAI and CESM the reader may need some more information concerning the CESM setup. Could you therefore include information on the native grid spacing of the CESM simulation and how this compares to ERAI? Also, it could be worthwhile to mention that the number of vertical levels is considerably higher in ERAI than in CESM.

The following information has been added to the text (see p.4, lines 11 ff)
The ERA-Interim Reanalysis which is used for assessing the ability of CESM to simulate WCBs in HIST, has a horizontal resolution of approx. 80 km and 60 vertical levels, thus twice as much as CESM1. The reduced number of vertical levels in CESM might have an influence on the calculation of WCB trajectories. Thus, the differences in the grid representation between ERA-Interim and CESM1 have to be kept in mind when comparing WCBs in ERA-Interim and CESM.

p. 4, l.12: I understand that the identification of WCBs in CESM is nearly identical to that in ERAI. Still, could you clearly mention possible differences (e.g., number of vertical levels).

At the end of section 2.1 we included a sentence mentioning explicitly the difference in the vertical levels between ERA-Interim and CESM1 (see comment above).

p. 5, l.14: If I recall correctly, ERAI WCB trajectories of Sprenger et al. (2017) were mapped to 1° grid spacing. What is the motivation for choosing 0.5° in CESM? Were the ERAI data

remapped to the same grid spacing prior to comparing the two data sets. If not, could the different resolution explain some of the differences seen in Fig. 2?

We added the following sentence about the gridding of both WCB data sets (see p.5, lines 17 ff).
This is done for WCBs calculated based on both datasets, ERA-Interim and CESM in order to make the gridded datasets comparable.

p. 6, l.2: I fully understand that the authors chose a period of 37 years from ERAI to have the largest possible sample size. Still I am wondering, if the authors calculated the ERAI WCB frequencies for the same period as with CESM (1990-1999) would the differences between the two datasets become even smaller?

This is an important point. There are, actually, different aspects to consider: First, by restricting the ERA-Interim calculation to the same period as in CESM, we would make the comparison period (10 years) of the two data sets equivalent, but would still have an imbalance of (then) ten years ERA-Interim vs. 50 years of CESM. We would, secondly, also restrict some of the external forcing included in the CESM simulation to correspond to the ERA-Interim forcing. This would apply, for instance, to the Pinatubo eruption in 1991. However, since CESM is a coupled atmosphere-ocean simulation, some important other forcing factors cannot be expected to match with observed forcings. For instance, the ENSO variability in CESM cannot be expected to match better with the observed variability.
In summary, even if we restrict the ERA-Interim period to 1990-1999, it is not immediately clear whether we can expect a better agreement between CESM and ERA-Interim, or not. On the other hand, extending the ERA-Interim period to 37 years, we expect to have a more robust estimation of the average WCB climatology, which also is more comparable to the 50-year climatology of CESM. The challenge and the limitations in comparing CESM and ERA-Interim WCB climatologies is discussed in Section 3 where we discuss the difference between ERA-Interim and HIST.
As suggested we also calculated the WCB climatology for the years 1990-1999 based on ERA-Interim and compared it to the CESM HIST climatology (see Figure 1, left panels, right panels show the 50 year CESM HIST climatology). Figure 1 is the same as Figure 2 in the manuscript  but the ERA-Interim climatology shows a 10 year mean from 1990-1999 only.

[Figure]

*Figure 1: Climatological frequency of WCBs for DJF (a, b) and JJA (c, d) for ERA-Interim for the years 1990-1999 (a, c) and HIST (b, d). Colours denote the percentage of six-hourly timesteps during which at least one WCB trajectory is located in a circle of 100\,km around the considered grid point, 24\,h after the start of the ascent. The grey line shows a frequency of 1\% for WCBs at their starting time t=0\,h and the red line shows a frequency of 1\% for WCBs at the end of their ascent at t=48\,h.*

The 10 year ERA-Interim climatology shows, as expected, a more noisy signal. The overall agreement between both datasets is still very good, however it did not improve by restricting the ERA-Interim period to the same 10 years (1990-1999) as have been used for the CESM simulations. We therefore decided to consider the full ERA-Interim time period for comparison to have a more robust estimation of the average WCB climatology.
We included additional text about the 10-year ERA-Interim climatology in section 3  (see p.8, lines 9ff).

p.6, l.28/29: I assume it should be Northwest Pacific instead of Northeast Pacific.

done

p. 6, l.32: The similarity between ERAI and CESM is indeed striking. Still, it seems to me that some differences may exist which are currently not seen due to the choice of figures. For example, the DJF WCB frequencies seem to be generally higher in ERAI than in CESM (Figs. 2a,b) and differences at t=0h and t=48h are hardly visible since only the 1% isoline is shown. Have the authors considered to include difference maps between the two? I would strongly encourage the authors to either include such difference maps in the main paper or at least in the supplemental material. For example, it would be interesting to see whether

biases in WCB frequency at t=48h correspond to biases in blocking frequency reported in previous studies (e.g., Woolings et al. 2018).

As suggested we include the difference maps ERA-Interim - HIST in the supplement for DJF (Figure S1) and JJA (Figure S2) and refer to them in the main text (see p.8, lines 3 ff).
The link to the blocking frequencies (Woolings et al., 2018) is also briefly discussed in line 5.

p. 8, l. 1: What exactly is shown in the intensity maps of the supplementary material? I would understand if the unit of the intensity maps was "number of trajectories/6h". Currently it is in %. Is this correct? If correct please explain how to understand the intensity in %.

Many thanks for pointing us to this unclarity. Indeed, we have to apologize for the unclear and, actually, misleading unit. The interesting comparison is, e.g., between Figure 4 in the main text and the corresponding Figure S1 in the Supplement Material. We describe now in detail in which way the two figures differ.
Figure 4c, for instance, considers how often a CESM grid point is located nearby a passing WCB trajectory, where 'nearby' means that the trajectory has to pass by within a 100 km distance and 'passing' refers to the time instance 24 h since the start of the WCB trajectory. The value is given as a percentage relative to all considered time instances. As an example: in the North West Pacific a WCB is passing by in 3-5% of all time instances.
Because Figure 4 provides 'only' frequencies, the information on how many WCB trajectories pass by is lost. If two WCB trajectories pass by within a 100 km distance, in the frequency map of Figure 4 it will only be considered that WCB trajectories passed by, but not how many did so. In contrast, as the reviewer correctly remarks, Figure S3, S4 of the Supplement Material provides the information on how many trajectories are passing by. Hence, Figure 4 shows frequencies , and Figure S3,S4 shows absolute numbers.
The correct unit of Figures S3 and S4 is '#trajectories / km2 6 h', what we refer to as a WCB intensity in the text. We corrected it in figures S3/S4 and in the revised manuscript.

p. 9, l.5/6: I guess it is 1990-1999 instead of 1990-2000.

done

p. 9, l.23: The decrease of WCB frequency at t=0,24, and 48h east of Madagascar is a rather stationary signal, i.e., the decrease in WCB outflow does not occur far downstream of WCB inflow as one would expect in midlatitudes. So, are the signals seen here really WCBs or is it rather the signature of recurving tropical cyclones that are expected to become less frequent in a warmer climate (Fig. 3 in Roberts et al. 2020)? This decrease of tropical cyclone frequency likely corresponds to the decrease in cyclone frequency seen in Fig. 7 of the present manuscript. Since the authors already tracked cyclones in CESM: would it be possible to determine the genesis region of cyclones around Madagascar? In my view, this would aid the interpretation of the results. Depending on the outcome of this analysis please consider to change the abstract and conclusion accordingly.

Many thanks for this valuable comment. Indeed, we fully agree with the reviewer that the 'WCB signal' around Madagascar asks for a  more careful discussion. We extended the discussion in 4.1 as suggested by the reviewer, i.e., by pointing out that the signal might

actually be related to recurving tropical cyclones. Thanks also for the reference to Roberts et al. (2020), which we also added to the discussion in the text.

The reviewer is also right in that it would be rather easy to deepen the discussion by exploiting the cyclone dataset we have at hand. Hence, we gratefully determined, as suggested, the genesis region of the Madagascar cyclones, and adapted the discussion, abstract and conclusions.

From Figure 7, a pronounced decrease of cyclones is seen in the southern Indian Ocean during austral summer (JJA). For convenience we reproduce this panel (Figure 7b) in an improved version:

[Figure]

Frequency of occurrence [%]

The marked decrease amounts to 10-15% during austral summer, and – as the reviewer points out – this might very well correspond to the diagnosed decrease in tropical cyclones by Roberts et al. (2020). Citing from Roberts et al. (2020):

> *Coupled models project a reduction of TC activity in the Southern Hemisphere, with the signal coming largely from changes in the Southern Indian Ocean and Australasian regions as also seen in CMIP5 (Bell et al., 2019; Gleixner et al., 2014; Tory et al., 2013).*

More quantitatively, we quantified the percentage decrease of cyclones in a box (lonmin=40, lonmax=75, latmin=25, latmax=-5) in RCP85 compared to HIST. We restricted this analysis to cyclones with a lifetime of at least 24 hours, and we also required that they experienced a deepening of at least 5 hPa. The number of cyclones is 195 in HIST and 153 in RCP, corresponding to a decrease of 22%.

Additionally, we required the cyclones to have genesis further east than their maximum intensity, i.e., we consider only westward moving cyclones and take this as a very simple proxy for tropical cyclones. It turns out that: 55.4 % of cyclones in HIST and 51.6 % of cyclones in RCP85 are westward moving (i.e., TCs), and that the percentage change of westward moving cyclones in RCP85 compared to HIST is -26.8 %. These numbers seem to

be consistent with Roberts et al. (2020), which also diagnoses comparable reductions in the Southern Indian Ocean.

The manuscript was revised in the following way: (i) Roberts et al. (2020) is added to the reference list and we discuss in the text the specific role of Madagascar cyclones and how the WCB signals in this region have to be interpreted; (ii) we remove the reference to the Madagascar WCBs in the abstract and conclusions as they reflect tropical cyclone air streams.

p. 9, l.32: To my understanding of Fig. 4f WCB outflow is less often located around 45°N rather than 30°N.

done

p. 10, l.1: The increase mostly occurs north of 60°N. Please correct.

done

p. 10, l.15: Also here, please double-check the units of Figures S1 and S2.

See reply to the previous comment. We clarified the distinction between WCB frequencies (in the main text) and WCB intensities (in the Supplement Material) amd changed the units in the supplement figure to  #trajectories/(km2 6h).

p. 10, l.33: Please double-check the latitude. I would rather say that the increase is found around 60°N.

done

p. 11, l.1: As mentioned above: The decrease of cyclone frequency around Madagascar could be a signal of changed tropical cyclone frequencies. Perhaps the authors can refer to the corresponding literature (e.g., Roberts et al. 2018) at this occasion.

We included the references and discussed the WCB occurrence in the vicinity of Madagaskar (see p.13, lines 17ff).

p. 13, footnote: "WCB air parcels no longer increase in altitude, i.e., the pressure increases again" might be slightly confusing. Better write "...no longer gain altitude..."?

done

p. 14, l.2: Is the increase in specific humidity what one would expect based on the Clausius-Clapeyron relation?

This is a good point. We consider, for instance, in Figure 8 the signals for the region NATL. The median of specific humidity increases from ~9.5 g/kg in present-day climate to ~12 g/kg in future climate. To relate this change in specific humidity to the Clausius-Clapeyron relation, the corresponding change in the temperature in the WCB inflow is needed. We

extracted this piece of information from the WCB data sets in the two CESM climate simulations and are able to see if the relation holds or not.

As a reference, we already cite here the Clausius-Clapeyron relationship according to the Glossary of Meteorology of the American Meteorological Society (glossary.ametsoc.org/ wiki/Clausius-clapeyron_equation):

$$e_s(T) = 0.6112 \exp\left(\frac{17.67\,T}{T + 243.5}\right),$$

Here, T is temperature in °C and $e_s(T)$ is the saturation vapor pressure in kPa. This formula is valid in the temperature range between -35°C < T < +35°C according to Bolton (1985). Using this approximate formula, one can show that at terrestrial conditions (T~260 K) a temperature change of 3 K results in a 20% change of the saturation vapor pressure. Assuming that the relative humidity remains fixed, the actual water vapor accordingly also increases by 20%. In summary, as a first approximation a 20% increase in specific humidity is expected for every 3 K temperature increase.

We have determined the temperature and the specific humidity at time t = 0 h (WCB inflow) for the HIST and RCP85 CESM simulations. They are listed in the following table for winter and separately for the four target regions (temperature in K, specific humidity in g/kg):

```
=============================================================
            | HIST            | RCP85           |
* * *
            | TEMP(0)  Q(0)  | TEMP(0)  Q(0)  | DT    DQ
----------|--------------------------------------------------
NATL (DJF) | 286.2    9.3   | 289.3   11.6  | 3.1   2.3 (19.8 %)
NPAC (DJF) | 286.4    9.5   | 288.9   11.5  | 2.5   2.0 (17.4 %)
SAM  (JJA) | 288.8    8.3   | 291.3   10.5  | 2.5   2.2 (21.0 %)
MED  (DJF) | 282.6    6.3   | 285.7    7.9  | 3.1   1.6 (20.3 %)
=============================================================
```

The results agree well with the expected 20% increase of specific humidity per 3 K temperature increase. For instance, in the region NATL temperature increases by 3.1 K (DT) and the corresponding change in specific humidity (DQ) is 2.3 g/kg, which amounts to a 19.8 % increase.

We mention in the revised manuscript in one sentence that the increases in specific humidity and temperature are in accordance with the Clausius-Clapeyron relationship (see p15, line 17), but refrain from discussing it in detail, i.e., we would prefer not to add the detailed analysis of the above table to the manuscript.

p. 14, l.8: I assume that the distinction between large-scale and convective precipitation is based on the classification by CESM (parametrized vs resolved). Please clarify in the text.

done

p. 17, l.6: I guess it is 5.0mm/48h instead of 50mm/48h.

done

p. 18, l.31: I understand that the maximum anomaly is located within the lower stratosphere. But is this also true when considering the absolute frequencies? Further, I am wondering would the increase in outflow frequency mean that transport across the (dynamic) tropopause increases? Have you considered to quantify the fraction of WCBs that eventually end up in the lower stratosphere based on their PV at t=48h. An increase of WCB outflow in the lower stratosphere would raise interesting questions concerning their contribution to stratospheric water vapour concentrations in a warming climate - a discussion which focuses often on deep convection (e.g., Smith et al. 2022).

Concerning the first part of the question about the location of the absolute frequencies:
The absolute frequencies of the location of WCB outflow is shown with the green lines in Figure 11. For all regions it can be seen that the maximum in the outflow location occurs at the dynamical tropopause for NATL and SATL and even slightly above the tropopause in NPAC, indicating that the WCB outflow has the potential to disturb the upper-level wave-guide and that this potential is increased in the RCP85 scenario because the outflow is shifted to higher altitudes (see colours).

Concerning Troposphere-Stratosphere exchange:
A simple (but still robust) way to determine the percentage of WCB trajectories reaching the stratosphere, is to determine for each WCB trajectory whether it has at the end of ascent, as defined in section 4.2 of the manuscript, a PV value above 2 PVU. We have done this for winter and separately for the four regions. The results for the HIST CESM and RCP85 simulations are listed in the following table:

```
==========================================
           | % STE (HIST) | % STE (RCP85)
-----------|------------------------------
NATL (DJF) | 1.13         | 1.35
NPAC (DJF) | 1.35         | 1.60
SAM  (JJA) | 0.43         | 0.59
MED  (DJF) | 0.81         | 0.68
==========================================
```

The percentage of WCB trajectories reaching the stratosphere at the end of ascent indeed increases, except for the region MED. However, the overall percentages are rather small, in the range of 1-2 %. Therefore, we would prefer not to discuss this topic in too great detail in the revised manuscript, but only to mention it briefly in one, two sentences.

Of course, the above analysis only considers stratosphere-troposphere exchange at the immediate WCB outflow, and it remains to be seen in a refined analysis whether at later times, and further downstream, the WCB air parcels are able to cross the tropopause, thus leading to enhanced percentages. This, however, goes beyond the scope of the present study, and would also not fit to Figure 11 which specifically looks at the end-of-ascent time.

In summary, we add one sentence to the manuscript (including a more general reference to stratosphere-troposphere exchange; Skerlak et al., 2014, see p. 20, lines 23 ff) where we point to potential implications of WCBs for STE and that they might change in a warming climate. But, in the same line, we mention that a refined analysis goes beyond the scope of this study.

Škerlak, B., Sprenger, M., and Wernli, H.: A global climatology of stratosphere–troposphere exchange using the ERA-Interim data set from 1979 to 2011, Atmos. Chem. Phys., 14, 913–937, https://doi.org/10.5194/acp-14-913-2014, 2014.

p. 20, l.2: Is the diabatic heating rate calculated explicitly or is it based on the potential temperature change along the trajectories? Please explain.

Many thanks for having identified this unclarity. Indeed, we determine the change in potential temperature along the trajectories as the diabatic heating rate. More specifically, we consider along the trajectories all six-hour time intervals and then determine for these intervals the change in potential temperature.
As an alternative, explicit calculation of the heating rate, we would need 3D Eulerian heating rates as CESM model output, which are not available in the model output.
We added that information to the text (see p.20, line 31).

p. 20, l.6: I agree that the anomaly suggests a poleward shift by 5°, but is this also true when considering absolute values (i.e., the sum of HIST and anomalies)?

We are not sure whether we fully understand this point. Possibly, the wording is not perfectly clear. The color shading does not correspond to an *anomaly*, but is the difference (RCP85-HIST) between the distribution of maximum heating for the two climate simulations. Hence, the green contour lines give the distribution for HIST, and the contour 'lines' for RCP85 are obtained by adding the color-shaded field to the green-contour field. We assume that the reviewer interpreted the figure in this way, although we would not refer to the color shading as an anomaly.
Figure 2 shows the absolute values for HIST and RCP85. Also in the absolute values an poleward and upward shift can be seen. Thus we keep the conclusions as in the original manuscript but we would prefer to not put this figure to the supplement because it does not provide new information.

[Figure]

*Figure 2: Absolute frequencies of the location of maximum diabatic heating for DJF for NATL (upper row) and NPAC (lower row) for HIST (left) and RCP85 (right). The figure is the same as Figure 12 in the manuscript but does not show the difference but the absolute values for HIST and RCP85.*

p. 21, l.30: Please include a brief description on how the percentage of total precipitation linked to WCBs is quantified (matching of WCB masks and precipitation fields I assume).

We added the following explanation to the manuscript (see p.22 lines 27ff).
In order to attribute the precipitation to WCBs, we mask all gridpoints which are part of the ascent phase of a WCB. More precisely, we select all longitude/latitude positions along the WCB trajectories, as long as the WCB is still in its ascent phase with pressure values larger than 400\,hPa. These positions are then interpolated to a regular grid. We thus obtain 2D-masks for every 6-hour timestep which contain all gridpoints which are part of an ascending WCB. The precipitation which occurs at these gridpoints is defined to be linked to WCBs.

p. 22, l.3: Is it really certain that the microphysical processes are reasonably captured in CESM? Perhaps this statement could be weakened by stating that the integrated effect of microphysical processes is reasonably captured.

done

p. 24, l.28: To my understanding it is not only the changes in WCB outflow which have the potential to disturb the jet stream but also the changes in WCB ascent. A poleward shift of the ascent regions will lead to an irrotational outflow closer to the upper-level jet. If the authors agree this aspect could be included in the discussion.

We included a short remake about the irrotational outflow in the text (see p.20, line 22).

p. 25, l.6: Remove colon after "in".

done

p. 25, l.22: In a recent study Steinfeld et al. (2022) showed that the frequency of WCBs in blocking anticyclones is expected to increase by 15%. Perhaps the authors could refer to their study.

We included the proposed reference (see p.26, line 34)

**Figures:**

Figs. 2, 4, 5, 7, 11, 12, 13: Please increase the font size of the axis labels.

All geographical maps have been improved in terms of projection, colorbars and figure labels. In Figure 11,12, all labels have been increased.

Figs. 11, 12: Please indicate that PV in the southern hemisphere has been multiplied by -1.

done

**References:**

Roberts, M.J., Camp, J., Seddon, J., Vidale, P.L., Hodges, K., Vannière, B., Mecking, J., Haarsma, R., Bellucci, A., Scoccimarro, E., Caron, L.-P., Chauvin, F., Terray, L., Valcke, S., Moine, M.-P., Putrasahan, D., Roberts, C.D., Senan, R., Zarzycki, C., Ullrich, P., Yamada, Y., Mizuta, R., Kodama, C., Fu, D., Zhang, Q., Danabasoglu, G., Rosenbloom, N., Wang, H. and Wu, L. (2020), Projected Future Changes in Tropical Cyclones Using the CMIP6 HighResMIP Multimodel Ensemble. Geophys. Res. Lett., 47: e2020GL088662. https://doi.org/10.1029/2020GL088662

Smith, J. W., Bushell, A. C., Butchart, N., Haynes, P. H., & Maycock, A. C. (2022). The effect of convective injection of ice on stratospheric water vapor in a changing climate. Geophysical Research Letters, 49, e2021GL097386. https://doi.org/10.1029/2021GL097386

Steinfeld, D., Sprenger, M., Beyerle, U., & Pfahl, S. (2022). Response of moist and dry processes in atmospheric blocking to climate change. Environmental Research Letters, 17(8), 084020. https://doi.org/10.1088/1748-9326/ac81af

Woollings, T., Barriopedro, D., Methven, J. et al. Blocking and its Response to Climate Change. Curr Clim Change Rep 4, 287–300 (2018). https://doi.org/10.1007/s40641-018-0108-z

**Reply to**

**Comment on wcd-2022-38**
Anonymous Referee #2
* * *
Referee comment on "Warm conveyor belts in present-day and future climate simulations. Part I: Climatology and impacts" by Hanna Joos et al., Weather Clim. Dynam. Discuss., https://doi.org/10.5194/wcd-2022-38-RC2, 2022
* * *
**Synopsis:**
In this study the authors investigate how the global frequency, intensity, and characteristics of warm conveyor belts (WCBs) are represented in a state-of-the-art climate model, and how they change in a high emission warming scenario. The study is thorough, considering not only the spatial distribution of the WCBs, but also the changing characteristics and how these relate to the key impacts of WCBs - namely precipitation and large-scale dynamics. I find the paper to be very well written, clear, and easy to read. I have only a few minor comments/edits.

Many thanks for this positive feedback. We are happy that the reviewer thinks that the results will be of interest to the *Weather and Climate Dynamics* readership and that it is well written, clear and easy to read. Still, we appreciate the few minor comments by the reviewer very much and will address them below point by point.

**Minor:**
Page 3, Lines 13-14: I think it's not clear yet what "corresponding signals" might refer to. I understand having read the paper, but when I first saw this I was unclear. Could this be reworded?

The sentence has been rewritten:
Is CESM, in present-day simulations, able to reasonably capture geographical patterns and seasonal frequencies of WCBs, as compared to the climatologies in ERA-Interim?

Data: Could you say something about the different resolution of the datasets, and whether this has an impact on the results? For example, does the "intensity" of the WCBs, i.e., the number of trajectories in a grid box, depend strongly on the resolution?

The following information has been added to the text (see p.4, lines 11ff).
The ERA-Interim Reanalysis which is used for assessing the ability of CESM to simulate WCBs in HIST, has a horizontal resolution of approx. 80 km and 60 vertical levels, thus twice as much as CESM1. The reduced number of vertical levels in CESM might have an influence on the calculation of WCB trajectories. Thus, the differences in the grid representation between ERA-Interim and CESM1 have to be kept in mind when comparing WCBs in ERA-Interim and CESM.

Additionally, we added the following sentence about the gridding of both WCB data sets (see p.5 lines 16 ff).

This is done for WCBs calculated based on both datasets, ERA-Interim and CESM in order to make the gridded datasets comparable.

Page 6, line 17: "chapter" should be "section".

Done